# Phenotypic impact of individual conserved neuronal microexons and their master regulators in zebrafish

Laura Lopez-Blanch[1,2], Cristina Rodríguez-Marin[1,2†], Federica Mantica[1,2†], Luis P Iñiguez[1,2], Jon Permanyer[1], Elizabeth M Kita[1], Tahnee Mackensen[1,2], Mireia Codina-Tobias[1‡], Francisco Romero-Ferrero[1,3], Jordi Fernandez-Albert[1], Myriam Cuadrado[4], Xosé R Bustelo[4,5], Gonzalo de Polavieja[3], Manuel Irimia[1,2,6]*

[1]Centre for Genomic Regulation (CRG), The Barcelona Institute of Science and Technology, Barcelona, Spain; [2]Department of Medicine and Life Sciences (MELIS), Universitat Pompeu Fabra, Barcelona, Spain; [3]Champalimaud Research, Champalimaud Foundation, Lisbon, Portugal; [4]Instituto de Biología Molecular y Celular del Cáncer, CSIC and Universidad de Salamanca, Salamanca, Spain; [5]Centro de Investigación Biomédica en Red de Cáncer (CIBERONC), Instituto de Salud Carlos III, Madrid, Spain; [6]ICREA, Barcelona, Spain

*For correspondence:
mirimia@gmail.com

†These authors contributed equally to this work

Present address: ‡Biozentrum, University of Basel, Basel, Switzerland

Competing interest: The authors declare that no competing interests exist.

## eLife Assessment

This **important** work examines how microexons contribute to brain activity, structure, and behavior. The authors find that loss of microexon sequences generally has subtle impacts on these metrics in larval zebrafish, with few exceptions. The evidence is **convincing**, using modern high-throughput phenotyping methodology in zebrafish. Overall, this work will be of interest to neuroscientists and generate further studies of interest to the field.

**Abstract** Microexons exhibit striking evolutionary conservation and are subject to precise, switch-like regulation in neurons, orchestrated by the splicing factors *Srrm3* and *Srrm4*. Disruption of these regulators in mice leads to severe neurological phenotypes, and their misregulation is linked to human disease. However, the specific microexons involved in these phenotypes and the effects of individual microexon deletions on neurodevelopment, physiology, and behavior remain poorly understood. To explore this, we generated zebrafish lines with deletions of 18 individual microexons, alongside *srrm3* and *srrm4* mutant lines, and conducted comprehensive phenotypic analyses. We discovered that while loss of *srrm3*, alone or together with *srrm4*, resulted in significant alterations in neuritogenesis, locomotion, and social behavior, individual microexon deletions typically produced mild or no noticeable effects. Nonetheless, we identified specific microexons associated with defects in neuritogenesis (*evi5b*, *vav2*, *itsn1*, *src*) and social behavior (*vti1a*, *kif1b*). Additionally, most micro-exon deletions triggered coordinated transcriptomic changes in neural pathways, suggesting the presence of molecular compensatory mechanisms. Our findings suggest that the severe phenotypes caused by *Srrm3/4* depletion arise from the combined effects of multiple subtle disruptions across various cellular pathways, which are individually well-tolerated.

## Introduction

Alternative splicing, the differential processing of introns and exons in eukaryotic pre-mRNAs, is the main mechanism generating transcriptomic and proteomic diversity in vertebrates. A large fraction of human multi-exonic genes exhibit alternative splicing (*Pan et al., 2008*; *Wang et al., 2008*), and a substantial portion of it is regulated in a cell-type- and tissue-dependent manner (*Tapial et al., 2017*). A paradigmatic example of tissue-specific regulation are neural microexons, tiny exons ranging from 3 to 27 nucleotides (nts) in length (*Irimia et al., 2014*). While a few microexons had been anecdotally reported before (e.g. *Worley et al., 1997*; *Dergai et al., 2010*), the discovery of hundreds of microexons regulated in a tight neural-specific manner dates back to a decade ago (*Irimia et al., 2014*; *Li et al., 2015*). Neural microexons are often completely skipped in most cell types, including stem cells and neural progenitors, and are sharply switched on during neuronal differentiation by their master regulators *Srrm3* and *Srrm4* (*Irimia et al., 2014*; *Nakano et al., 2019*). More recently, inclusion of a subset of highly sensitive neural microexons has been reported in endocrine pancreatic cells of mammals, where *Srrm3* is also expressed (*Juan-Mateu et al., 2023*; *Bonnal et al., 2025*).

The potential functional relevance of cell-type-specific microexons is supported by multiple lines of evidence. First, neural microexons are the most conserved type of alternative splicing described so far, showing remarkable conservation across jawed vertebrates and, in some instances, from human to cephalochordates and arthropods (*Torres-Méndez et al., 2019*; *Torres-Méndez et al., 2022*). Second, the impact of numerous microexons on protein function has been demonstrated, and they often modulate protein-protein interactions or protein interactions with other molecules (*Ellis et al., 2012*; *Kjer-Hansen and Weatheritt, 2023*; *Roth et al., 2023*). Third, loss-of-function models for their highly specific master regulators show dramatic phenotypes, particularly in mammals. For instance, a homozygous mouse gene trap for *Srrm4* exhibits strong neurodevelopmental defects (*Quesnel-Vallières et al., 2015*), while another gene trap for *Srrm3* also displays multiple neurodevelopmental phenotypes (*Nakano et al., 2019*), as well as pancreas-derived physiological alterations (*Juan-Mateu et al., 2023*). Moreover, the heterozygous mouse gene trap for *Srrm4* shows defects on neuronal activity and it exhibits multiple hallmarks of autism spectrum disorder (ASD; *Quesnel-Vallières et al., 2016*), in line with the reported misregulation of microexons in various human patients with ASD (*Irimia et al., 2014*). In zebrafish, we have recently demonstrated that *srrm3* mutants exhibit severe photoreceptor malformations and degeneration, leading to vision loss and early mortality under standard rearing conditions (*Ciampi et al., 2022*).

Despite these studies, however, how misregulation of specific microexons leads to the dramatic phenotypes observed for the mutants of their master regulators is still largely unknown. For instance, it is not yet known whether these phenotypes are caused by a few microexon targets with strong effects (e.g. as in the paradigmatic example of FGFR alternative splicing for *Esrp1/2 Bebee et al., 2015*; *Burguera et al., 2017*) or by a complex cumulative effect of multiple small molecular impacts. A major limitation is that the cellular and organismal phenotypes caused by the removal or downregulation of individual microexons has only been assessed for a few cases, mainly using mouse models or cell cultures. In many such cases, loss-of-function microexon models show relatively mild but robust phenotypes. Importantly, these phenotypes range from defects on neurite growth (*Worley et al., 1997*; *Zibetti et al., 2010*; *Ohnishi et al., 2014*; *Quesnel-Vallières et al., 2015*) to neuronal malfunction and behavioral abnormalities (*Wang et al., 2015*; *Gonatopoulos-Pournatzis et al., 2020*; *Poliński et al., 2025b*). However, other cases did not reveal noticeable phenotypes (e.g., *Matalkah et al., 2022*), and the potential impact of the publication bias towards positive results is difficult to evaluate.

To tackle this major open question, common to other tissue-specific splicing regulators, in this study we generated stable CRISPR-Cas9 deletion mutants in zebrafish for 18 highly conserved neural-specific microexons regulated by *srrm3/4*. We phenotypically characterized these lines, along with mutant lines for *srrm3* and *srrm4*, across multiple levels, including axon guidance, neurite growth, larval and juvenile locomotion, larval sensory responses, and social behavior in juveniles. In contrast to the marked phenotypes observed for their master regulators, especially *srrm3*, most microexon mutants showed no phenotypes for most tests, with a few notable exceptions. Transcriptomic analysis of mutant larvae and their wild type (WT) siblings suggests that coordinated (mis)regulation of specific neural-related pathways may provide functional compensations at the molecular level. In summary, while the presence of non-studied microexon targets with strong functional effects cannot be ruled

out, our results are in line with a model in which the phenotypes of the master regulators are caused by the combined impact of multiple milder perturbations affecting many cellular pathways.

## Results

### Identification and characterization of *srrm3/4*-regulated neural microexons in zebrafish

To identify neural exons and microexons in zebrafish, we employed a similar approach to that described recently for mammals (*Juan-Mateu et al., 2023*; see Methods). Using transcriptome-wide quantifications of exon inclusion for multiple cell type and tissue samples from *VastDB* (*Tapial et al., 2017*), we identified a total of 708 neural exons with sufficient read coverage in at least five tissues. Of these, 246 (34.8%) corresponded to microexons of 3–27 nt, 139 (19.6%) to 28–51 nt exons and 323 (45.6%) to exons longer than 51 nts (*Figure 1—figure supplement 1A*, *Supplementary file 1*). For comparison, we identified 919 neural exons in human with equivalent expression criteria, which showed a similar proportion of microexons (*Figure 1—figure supplement 1A*). As in the case of mammalian programs (*Juan-Mateu et al., 2023*), a subset of zebrafish neural microexons were also substantially included in endocrine pancreatic cells, whereas smaller fractions were detected in muscle and, less expectedly, immune tissues (*Figure 1A*). Tissue-specific regulation was validated by RT-PCR assays for all 21 selected conserved neural microexons (see below) using independent RNA samples (*Figure 1—figure supplement 1B*).

Neural microexons show sharp progressive upregulation during embryo development (*Figure 1B*), in line with an increase in neuronal differentiation. In terms of function, genes harboring microexons are strongly enriched in Gene Ontology (GO) terms related to vesicle-mediated transport, endocytosis, GTPase activity, synaptic organization and neuron development (*Figure 1C*), consistent with previous studies (*Irimia et al., 2014*; *Torres-Méndez et al., 2022*). Zebrafish microexons are also significantly more conserved in human than long exons in terms of neural regulation (*Figure 1D*; p=6.8e-03, two-sided Fisher's Exact test between microexons and long exons). However, this increased conservation was relatively minor and, in fact, similar at the genomic level ('no event ortholog', i.e. no homologs in humans, in *Figure 1D*). This contrasts with the reciprocal comparison, where human neural microexons appeared substantially more conserved at the genomic and regulatory level in zebrafish (*Figure 1—figure supplement 1C*), and suggests that zebrafish has evolved a larger fraction of lineage-specific microexons. Finally, 191/243 (78.6%) of neural microexons with sufficient RNA-seq read coverage in mutants and control siblings were affected (ΔPSI < –15) by *srrm3/4* loss-of-function in 5 dpf larva and/or retina, in contrast to 82/136 (60.3%) and 122/317 (38.5%) exons of length 28–51 nts and longer, respectively (*Figure 1E*; p=8.4e-22, two-sided Fisher's Exact test between microexons and long exons; see Materials and methods). A similar fraction of human orthologs (98/121, 81.0%) responded to the ectopic expression of the regulator in HEK293 cells (*Figure 1—figure supplement 1D*). In addition, the genome-wide effect of *srrm3/4* loss-of-function in zebrafish is strongly biased towards small exons (*Figure 1F*), as we previously reported (*Ciampi et al., 2022*).

### Stable microexon deletions do not compromise viability and overall morphology

To investigate the functional impact of neural-specific *srrm3/4*-regulated microexons, we selected 21 highly conserved microexons for further characterization (*Supplementary file 2*). These microexons were selected based on various criteria, including sharp neural-specific inclusion and strong *srrm3/4* regulation in zebrafish, as well as high regulatory conservation in human, both in terms of tissue-specificity and *SRRM3/4* regulation (*Figure 2A*). In addition, we prioritized some microexons for which a molecular function had been reported (e.g. *apbb1*, *itsn1*, *kif1b*, *mef2ca*, *src*), and the host gene had no close paralogs (e.g. *reln*, *madd*, *src*) and/or had important roles in development and neurobiology (*gli2b*, *gli3*, *cdon*, *mef2ca*, *shank3a*, *shank3b*, *reln*). Moreover, we included representatives of the main enriched GO categories (*Figure 1C*), such as vesicle transport and endocytosis (*itsn1*, *src*, *vti1a*), GTPase activity (*vav2*, *evi5b*, *ap1g1*, *asap1b*, *madd*, *dock7*), synapse organization (*shank3a*, *shank3b*), and microtubule binding (*kif1b*).

We generated stable zebrafish lines for the deletion of each of these microexons using the CRISPR-Cas9 system by designing pairs of guide RNAs (gRNAs) targeting the neighboring upstream and

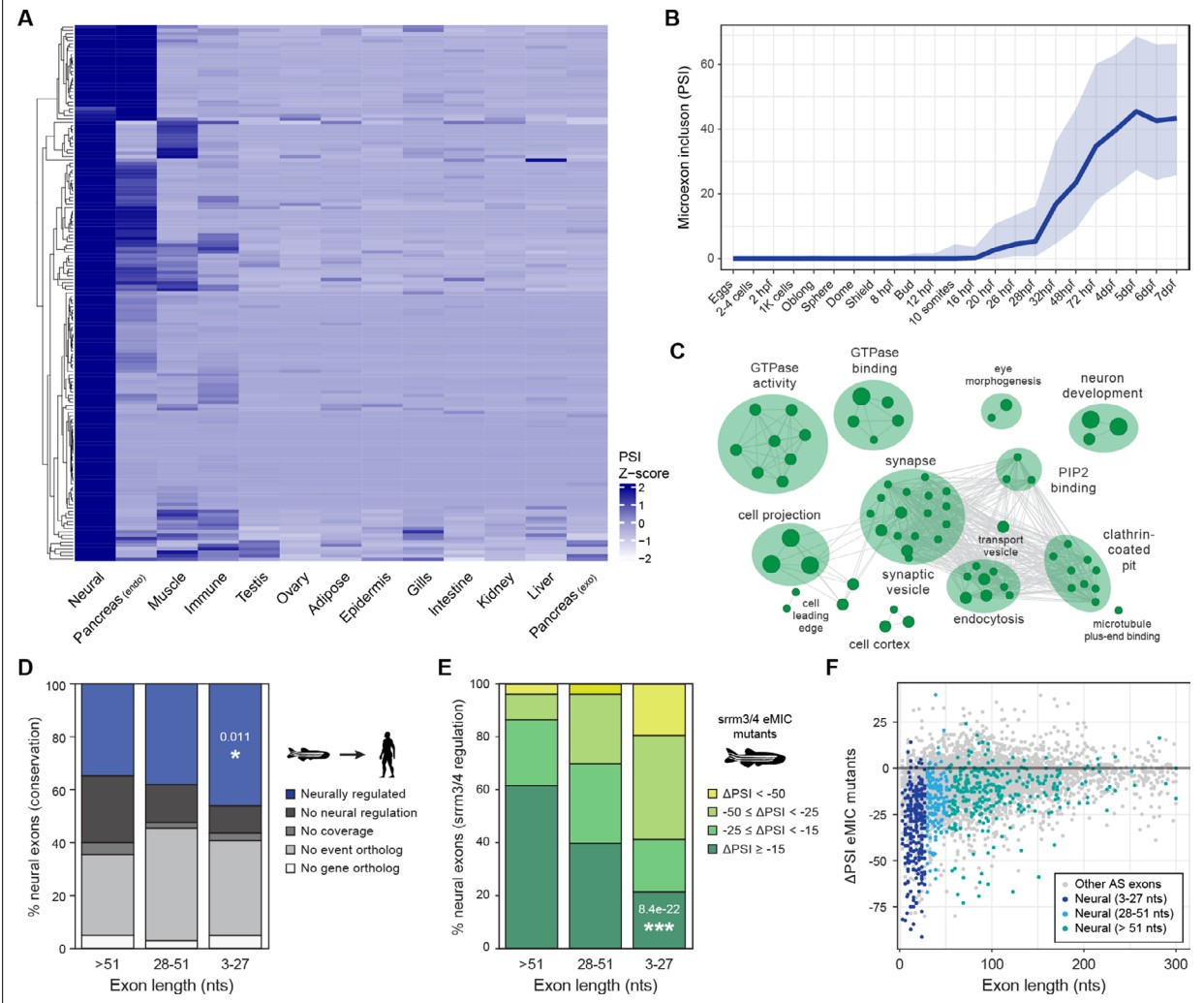

**Figure 1.** Neural microexon program in zebrafish. (**A**) Heatmap showing relative inclusion levels of neural microexons identified in zebrafish (**Supplementary file 1**). The value for each tissue type corresponds to the average inclusion in the samples for that tissue in *VastDB* (**Supplementary file 1**). Only events with sufficient read coverage in >10 tissue groups are plotted (N=157) and missing values were imputed. (**B**) Inclusion of neural microexons along embryo development (egg to 7 days post-fertilization [dpf]). Thick line corresponds to the median PSI and the shades to the first and third quartile of the PSI distribution. (**C**) Enriched Gene Ontology (GO) categories among genes harboring neural microexons in zebrafish. GO terms were grouped into networks by ClueGO. (**D**) Evolutionary conservation of zebrafish neural exons of different length group at the genomic and tissue-regulatory level compared with human. Exons conserved at the regulatory level (blue) are those with enriched inclusion in neural samples (ΔPSI ≥15) also in human. Those with no neural regulation (black) are conserved at the genomic level (*Irimia et al., 2009*), but are not neurally enriched. Those conserved but with insufficient coverage to assess regulation are indicated as "No coverage" (dark grey). P-value corresponds to a two-sided Fisher's Exact test for neural conservation vs. others between exons >51 nts and 3–27 nts. (**E**) Distribution of exons of different length by the level of *srrm3/4* misregulation in larva or retina (see Methods). P-value corresponds to a two-sided Fisher's Exact test for non-regulated exons (ΔPSI ≥ −15) vs. others between exons >51 nts and 3–27 nts. (**F**) Change in inclusion levels [ΔPSI (enhancer of microexons domain (eMIC) mutant-WT)] for all exons shorter than 300 bp. Dots with different blue colors correspond to neural exons of different length.

The online version of this article includes the following figure supplement(s) for figure 1:

**Figure supplement 1.** Identification and validation of neural microexons in zebrafish.

downstream intronic sequence (**Figure 2B**, **Supplementary file 3**; see Materials and methods). We selected two founders for each microexon showing a clean genomic deletion covering the entire microexon. Such deletions are expected to result in normal expression of the host gene but skipping the microexon in every cell, including neurons, which can be validated through RT-PCR assays with primers spanning the neighboring exons (**Figure 2B, C**, **Figure 2—figure supplement 1**). However, for three of these microexons (*spock3*, *pus7*, and *dock7*; asterisks in **Figure 2A**), the genomic deletion

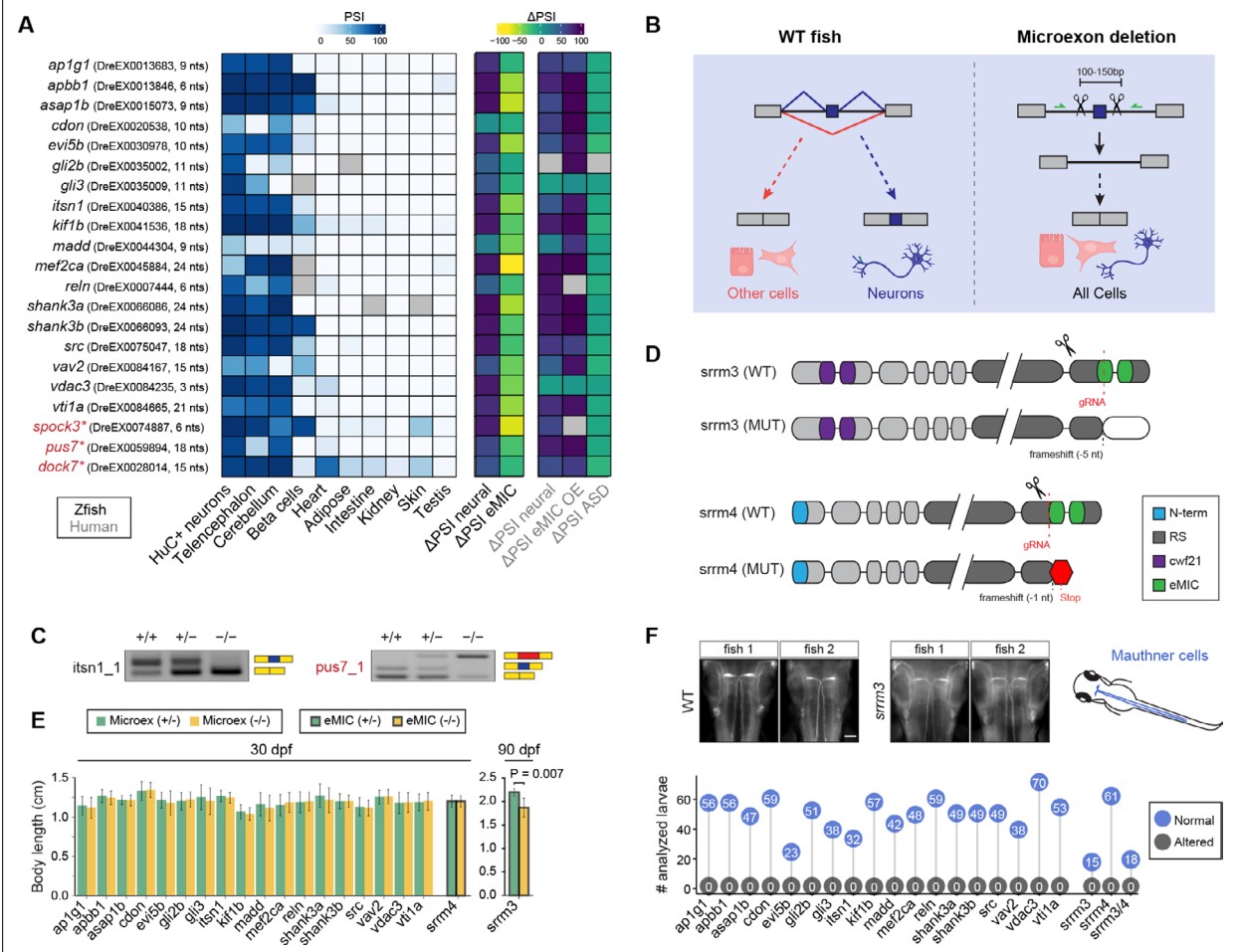

**Figure 2.** Selected neural microexons and experimental design. (**A**) Inclusion levels of the 21 selected conserved microexons across zebrafish tissues as well as differential inclusion in neural vs other tissues (ΔPSI neural) in zebrafish and human, change in inclusion in response to eMIC depletion in zebrafish (ΔPSI eMIC) or eMIC overexpression in human (ΔPSI eMIC OE), and change in inclusion between ASD patients and control individuals (ΔPSI ASD). The three microexons with asterisks were excluded from phenotypic analyses. (**B**) Schematic representation of the CRISPR-Cas9 based deletions of individual microexons. A pair of guide RNAs flanking each microexon were designed, which is expected to lead to normal gene expression without the microexon in all cells. (**C**) Examples of two RT-PCRs testing the inclusion of the targeted microexon upon CRISPR-Cas9 removal. *itsn1_1* shows the expected clean deletion in the homozygous, while *pus7_1* exhibits inclusion of a cryptic sequence of higher length (red block) (see *Figure 2—figure supplement 1*). (**D**) Schematic representation of *srrm3* and *srrm4* protein domains and the impact of the CRISPR-Cas9 derived mutations (from *Ciampi et al., 2022*). (**E**) Distribution of body lengths at 30 dpf for heterozygous (green) and homozygous (yellow) fish for each microexon deletion or the *srrm4* mutation. For *srrm3*, the values are shown for 90 dpf. P-value corresponds to a two-sided t-test. Error bars correspond to standard errors. (**F**) Top: two representative images from WT and *srrm3* homozygous mutant larvae showing staining for 3A10 in Mauthner cells (schematized on the right side). Bottom: quantification of normal and altered number of larvae with respect to Mauthner cell morphology in homozygous mutants for each microexon or regulator line.

The online version of this article includes the following figure supplement(s) for figure 2:

**Figure supplement 1.** Validation of microexon deletion lines.

led to the inclusion of cryptic sequences predicted to create frameshifts and thus gene-level loss of function, and were consequently excluded from subsequent phenotypic analyses. For comparison, we also included mutant lines for the master regulators of microexons, *srrm3* and *srrm4*, as well as the double mutant line (*Figure 2D*). Our previous work on these lines (*Ciampi et al., 2022*) showed that mutation of *srrm3* has a major impact on microexon inclusion in the retina, whereas *srrm4* loss of function has a very small effect.

Initial characterization of the 18 microexon deletions revealed no gross morphological abnormalities and no differences in body size compared to their heterozygous siblings at the juvenile stage (~30 dpf, *Figure 2E*). Similar results were obtained for *srrm4* mutants. This contrasts with the *srrm3*

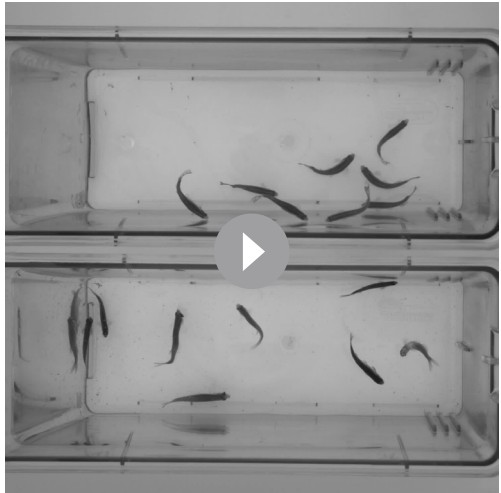

**Video 1.** Behavior of adult srrm3 mutants. Also accessible at https://data.mendeley.com/datasets/3b3zx4cfg9/1.
https://elifesciences.org/articles/104275/figures#video1

mutant line, for which we had previously shown that homozygous mutants die between 12 and 20 dpf, likely due to feeding problems caused by their impaired vision (*Ciampi et al., 2022*). Here, we have reared these fish using special conditions that increase food availability, allowing them to reach the adult stage (see Materials and methods). However, adult *srrm3* homozygous mutants were subfertile, showed reduced body size at 90 dpf (*Figure 2E*; p=0.007, two-sided t-test) and had multiple locomotor and behavioral abnormalities (*Video 1* and below).

## Some microexons exhibit robust defects on neurite outgrowth ex vivo

To begin investigating the impact of microexon deletions on zebrafish neurodevelopment, we assessed two phenotypes at the cellular level related to axon guidance and neurite outgrowth. Homozygous *Srrm4* KO mice showed severe defects on axonal midline crossing through the corpus callosum (*Quesnel-Vallières et al., 2015*). Thus, to evaluate potential axon guidance alterations, we first studied Mauthner cells in individual 48 hpf larvae. All studied larvae for all microexon deletions appeared normal, as well as those from *srrm3*, *srrm4* and double mutant lines (*Figure 2F*).

Next, we focused on neurite outgrowth, a phenotypic trait reported to be altered in primary neuronal cultures of homozygous *Srrm4* KO mice (*Quesnel-Vallières et al., 2015*), as well as for various loss-of-function models of individual microexons in mouse (e.g. *Kdm1a Zibetti et al., 2010*, *Zfyve27 Ohnishi et al., 2014*; *Ohnishi et al., 2017*, *L1cam Jacob et al., 2002*). To robustly quantify neurite growth, we developed a new protocol to generate primary neuronal cultures from zebrafish larvae (*Figure 3—figure supplement 1A*, see Materials and methods). In brief, HuC:GFP-positive cells were isolated using Fluorescence-activated Cell Sorting (FACS) at 48 hpf and plated on Poly-D-Lysine coated 8-microwell glass slides (1 cm$^2$/well) at a density of 200,000 cells/well in Neurobasal (NB) supplemented media at 28 °C and 5% $CO_2$. Isolated HuC:GFP-positive cells lose their neurites, which are then progressively regrown in culture, as observed by acetylated tubulin staining (*Figure 3—figure supplement 1B and C*). For each microexon deletion, as well as the regulator mutant lines, we measured the longest neurite for each cell at 10 and 24 hours after plating (hap; *Figure 3A*). A few microexon deletion mutants showed significantly different neurite length distributions compared to control WT larvae at 10 hap and/or 24 hap that were highly robust across replicates and founder lines (*Figure 3B-D*, *Figure 3—figure supplement 2*). Interestingly, deletions resulted in opposite differences in overall neurite length depending on the microexon: decreased (*vav2*), increased (*evi5b*, *itsn1*) or with distinct effects at 10 and 24 hap (*src*). In the case of the regulators, whereas neither *srrm3* nor *srrm4* individual mutants exhibited significant alterations in neurite length, the double mutant had strongly decreased neurite lengths at 10 hap compared to control siblings (*Figure 3B-D*, *Figure 3—figure supplement 2*).

At the molecular level, the microexons in both ITSN1 and SRC proteins have been shown to modulate interaction with DNM1 (*Dergai et al., 2010*), which is involved in synaptic vesicle recycling and is important for neurite outgrowth (*Bodmer et al., 2011*). In the case of the 10-nt microexon in *evi5b*, the microexons falls at the 3′ of the gene and its inclusion is predicted to generate an alternative C-termini (*Figure 3E*). Since *evi5b* is a GTPase activating protein (GAP) of *rab11* and overexpression of *rab11* has been reported to increase neurite outgrowth (*Eva et al., 2010*), we tested whether microexon inclusion altered its interaction with RAB11. However, protein-protein NanoBRET assays revealed a strong, but similar, interaction with RAB11 for both EVI5B isoforms (*Figure 3F*). Finally, in *vav2*, which encodes a Guanine nucleotide exchange factor (GEF), the deleted 15-nt microexon falls

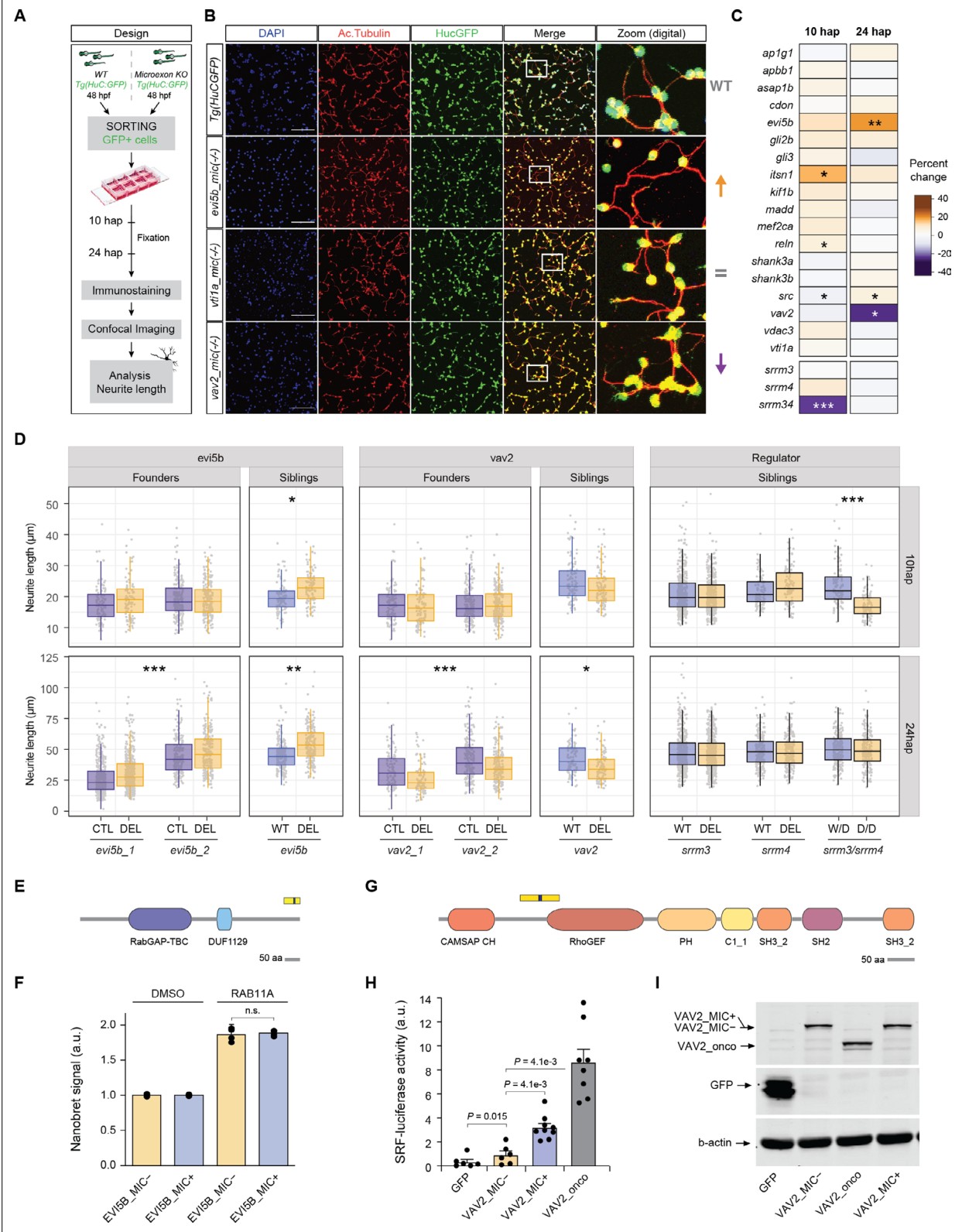

**Figure 3.** Impact of microexon deletion on neurite outgrowth. (**A**) Schematic representation of the experimental design used to assess neurite outgrowth in zebrafish neuronal primary cultures (see Methods). (**B**) Confocal images of example microexon deletions at 24 hours post-plating (hap). ×20 magnification images, white squares indicate the zoom region amplified in the right panel (digital zoom). Scale bar 50 μm. (**C**) Heatmap showing the median percent of change in neurite length at 10 and 24 hap of the homozygous mutant with respect to the matched control neurons (HuC:GFP

*Figure 3 continued on next page*

*Figure 3 continued*

line) for each main microexon deletion line (data for all tested lines in *Figure 3—figure supplement 2*). Significance is based on the median of p-value distribution of 10,000 bootstrap resampling Wilcoxon tests for each main founder. * 0.01<p ≤ 0.001, ** 0.001<p ≤ 0.0001, *** p<0.0001. (**D**) Boxplots of the distribution of the length of the longest neurite (one neuron per data point) for both founder lines generated for either *evi5b* or *vav2* (_1 and _2), as well as the length distributions for neurons of siblings that are either WT and homozygous deletion (Del) for the main founder lines (*evi5b_1* and *vav2_1*). For the regulators, neurite length distributions for neurons of WT and homozygous mutants (Del) siblings are shown for *srrm3*, *srrm4* and the double mutant *srrm3/4*, for which the control sibling corresponds to the WT of *srrm3* but homozygous mutant for *srrm4* (i.e., *srrm4* Del). p-Values correspond to ANOVA tests for *evi5b* and *vav2* founders and for siblings the median of 10,000 bootstrap Wilcoxon tests. * 0.01<p ≤ 0.001, ** 0.001<p ≤ 0.0001, *** p<0.0001. (**E**) Schematic representation of the domain architecture of EVI5B in zebrafish and region encoded by the microexon and upstream and downstream exons (upper blocks). Microexon inclusion/exclusion leads to different C-termini. (**F**) NanoBRET quantification of protein-protein interaction between EVI5B with and without the microexon and RAB11 proteins from zebrafish. Error bars correspond to standard errors. (**G**) Schematic representation of the domain architecture of VAV2 in zebrafish and region encoded by the microexon and upstream and downstream exons (upper blocks). (**H**) Luciferase readout of the activation of SRF upon ectopic expression of zebrafish VAV2 proteins with and without the microexon, a negative control and an oncogenic VAV2 mutant protein in COS7 cells. P-values correspond to Student's t-test after Holm-Sidak's multiple test correction. Error bars correspond to standard errors. (**I**) Western blot for each overexpressed protein tagged to GFP, detected by anti-GFP antibody. Bottom blot: β-actin is shown as loading control.

The online version of this article includes the following source data and figure supplement(s) for figure 3:

**Source data 1.** Unedited uncropped western blots with bands highlighted.

**Source data 2.** Raw, uncropped western blot gel images displayed in *Figure 3I* using an anti-GFP or an anti-β-actin antibody.

**Figure supplement 1.** Zebrafish neuronal primary culture.

**Figure supplement 2.** Quantification of neurite outgrowth across zebrafish mutant lines.

close to its RhoGEF domain (*Figure 3G*). To test if it could affect its GEF activity, we performed a SRF promoter activation assay upon *vav2* overexpression in COS cells, a well-established method to assess VAV protein activity (see Materials and methods). Whereas overexpression of both zebrafish VAV2 isoforms, with and without the microexon, increased reporter activation with respect to the negative control, the microexon-containing isoform displayed significantly stronger activity (*Figure 3H, I*; ~threefold increase over the skipped form, p=4.1e-03, Student's t-test). This increase in activity, however, was lower than that found in cells expressing a constitutively active, oncogenic version of VAV2 (*Figure 3H, I*). The catalytic activity of this mutant version of VAV2 is fully deregulated due to the elimination of the autoinhibitory N-terminal domains of VAV2, which leads to the generation of protein version that does not require tyrosine phosphorylation to activate it catalytic activity (*Schuebel et al., 1998*; *Bustelo, 2014*).

## Individual microexon deletions have little impact on larval activity patterns

We next assessed potential defects in larval activity levels, thigmotaxis and habituation to mechanical tapping stimuli. For this purpose, for each line, we placed 5 dpf larvae derived from a cross of two heterozygous parents into individual wells of a 48-well plate and recorded their activity using a DanioVision system during a predefined series of intervals under light or dark conditions (*Figure 4A*; see Materials and methods). Specifically, after 5 min of habituation, we recorded baseline activity in the light for 25 min, and then for five alternating intervals of 10 min of dark and light. Finally, a series of 30 tappings (1 Hz frequency) was applied. Overall, we found that homozygous *srrm3* and *srrm3/4* mutant lines showed significant and strong alterations for multiple parameters compared to their WT and heterozygous siblings (*Figure 4*, *Figure 4—figure supplements 1–13*). In contrast, *srrm4* and microexon deletion lines exhibited little or no defects, and those were generally not consistent across clutches and founder lines (*Figure 4—figure supplements 1–13*). In particular, *srrm3* mutants had increased activity in baseline and light interval conditions (*Figure 4C*), whereas the double mutant showed only a significant, yet milder, increase during the light intervals. Interestingly, larvae with homozygous *evi5b* microexon deletion also showed mildly significantly increased baseline activity (*Figure 4C*). In the case of the transitions between dark and light conditions, and vice versa, *srrm3* and *srrm3/4* mutants had strong and significant alterations in the relative activity response compared to control siblings (*Figure 4B, D*, *Figure 4—figure supplements 5 and 6*). This phenotype could be in part due to the impaired vision of these mutants (*Ciampi et al., 2022*), although they nonetheless showed a substantial response to changes in the light/dark conditions (*Figure 4—figure supplement*

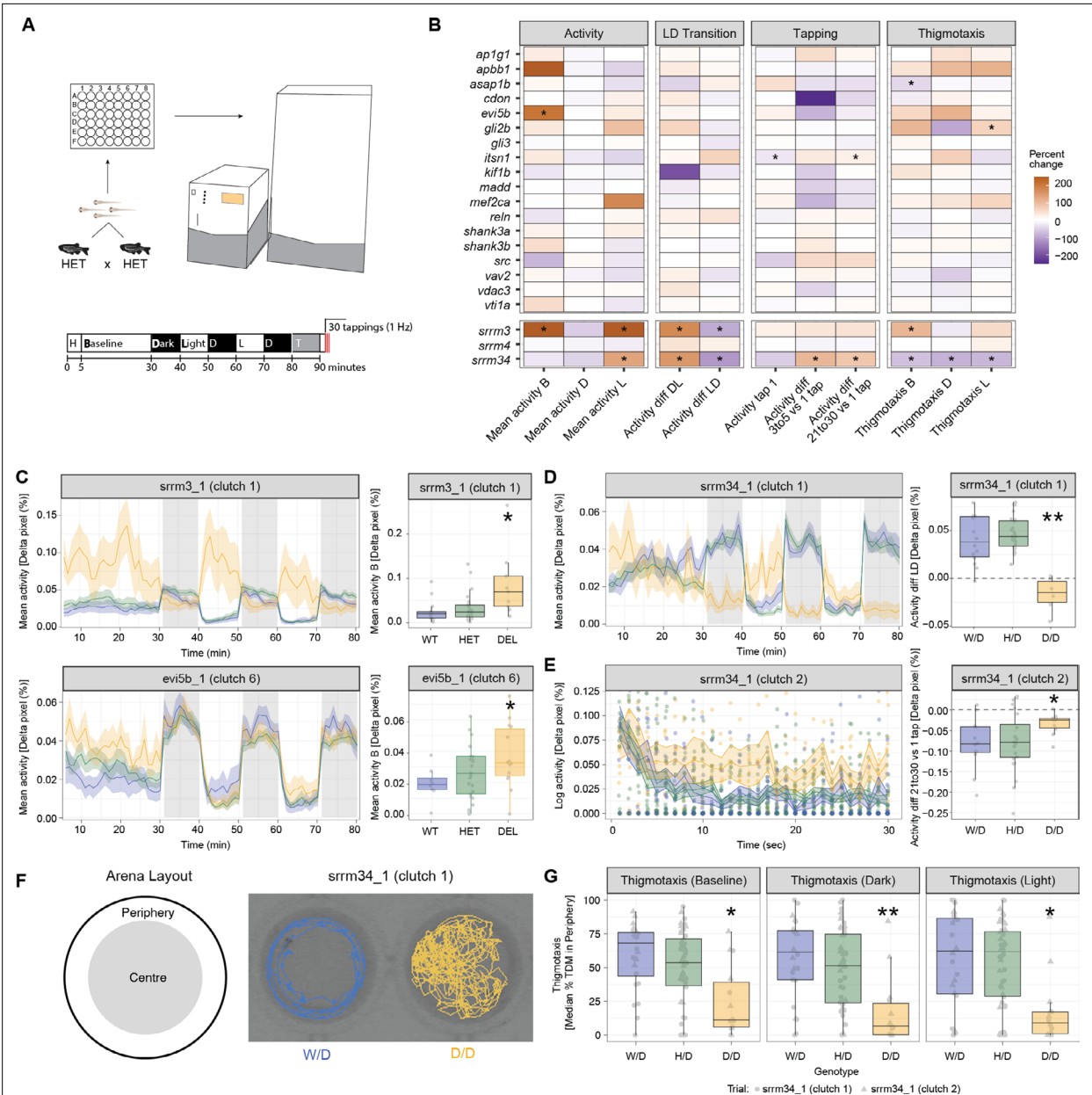

**Figure 4.** Impact of microexon misregulation in larval activity and response to stimuli. (**A**) Experimental design to assess larval activity patterns and response to stimuli using the DanioVision system and the EthoVision XT - Video tracking software. The protocol we implemented consists of 5' of habituation, 25' of baseline recording, five alternating 10' intervals of dark-light, 10' of re-habituation and the tapping experiment (30 taps at 1 Hz). (**B**) Heatmap showing the percent of change with respect to the WT value for the homozygous (Del) of each main microexon deletion line as well as the single and double regulator mutants for features related to activity and response to stimuli (full plots in *Figure 4—figure supplements 1–13*). For visualization purposes, the percent change for median WT values equal to zero was computed using the minimum median WT value of the relative category across founders. p-Values correspond to the median p-value from 100 permutation tests selecting 10 observations per genotype across replicates of the main founder. (**C**) Left: Activity (percentage of Δpixels/min) plots for a representative clutch of *srrm3* and *evi5b* main lines. Traces represent mean ± SEM across larvae from the same clutch and genotype. Dark and light periods are shown with gray or white background, respectively. Right: boxplots showing the distribution of the mean baseline activity for each larva of each genotype. p-Values correspond to two-sided Wilcoxon Rank-Sum tests of each genotype against the WT. (**D**) Left: Activity (percentage of Δpixels/min) plots for a representative clutch of *srrm3/4* double mutant line. Traces represent mean ± SEM across larvae from the same clutch and genotype. Dark and light periods are shown with gray or white background, respectively. Right: boxplots showing the mean difference in activity during the dark-to-light transition for each larva of each genotype. p-Values correspond to two-sided Wilcoxon Rank-Sum tests. (**E**) Left: Activity (percentage of Δpixels/s) after each of the 30 consecutive taps representative clutch of *srrm3/4* double mutant line. Traces represent mean ± SEM across larvae from the same clutch and genotype. (**F**) Left: schematic representation of a well, divided between center and periphery regions. Right: two representative tracks of WT and Del *srrm3/4* larvae for 60 s at baseline. (**G**) Boxplots

*Figure 4 continued on next page*

*Figure 4 continued*

showing the median percentage total distance moved (TDM) (mm) in the periphery for larvae of different genotypes of the *srrm3/4* double mutant line, under baseline, dark and light conditions. p-Values correspond to two-sided Wilcoxon Rank-Sum tests of each genotype against the WT. In D-G, W/D denotes *srrm3+/+,srrm4-/-* fish, H/D *srrm3+/-,srrm4-/-*, and D/D *srrm3-/-,srrm4-/-*. For all panels: * $0.05 < p \leq 0.001$, ** $0.001 < p \leq 0.0001$, *** $p < 0.0001$.

The online version of this article includes the following figure supplement(s) for figure 4:

**Figure supplement 1.** Larval activity over the time course of the experiment.

**Figure supplement 2.** Activity response and habituation to tapping stimuli.

**Figure supplement 3.** Baseline activity.

**Figure supplement 4.** Activity during the light intervals.

**Figure supplement 5.** Activity during the dark intervals.

**Figure supplement 6.** Activity during the dark-to-light transitions.

**Figure supplement 7.** Activity during the light-to-dark transitions.

**Figure supplement 8.** Median activity after the first tap.

**Figure supplement 9.** Difference in activity between the first tap and taps 3–5.

**Figure supplement 10.** Difference in activity between the first tap and taps 21–30.

**Figure supplement 11.** Thigmotaxis during the baseline condition.

**Figure supplement 12.** Thigmotaxis during the light intervals.

**Figure supplement 13.** Thigmotaxis during the dark intervals.

*1*). In contrast, only double *srrm3/4* mutant larvae displayed significant alterations in habituation to tapping, measured as the difference in activity between the first tap and taps 3–5 or taps 21–30 (*Figure 4E*). Finally, we also evaluated thigmotaxis, or the preference of a larva to be close to the wall of the well (*Figure 4F*), since increased time spent in the periphery is generally associated with higher anxiety levels in multiple systems and model organisms. Surprisingly, both *srrm3* and *srrm3/4* mutants showed significant abnormalities in thigmotaxis, but they did so in opposite directions (*Figure 4B*). Where *srrm3* homozygous mutant fish exhibited a bias towards the periphery in baseline conditions, double *srrm3/4* mutants had a robust and consistent preference for the center of the well, which was significant for baseline, dark and light intervals (*Figure 4F and G*). In summary, these studies show that, whereas globally misregulating microexons has a major impact on multiple activity and stimuli response patterns, deleting individual microexons has only a minor impact on those patterns.

## Effects of global and individual microexon misregulation on juvenile social behavior

Neural microexons were found to be misregulated in the brains of some individuals with ASD (*Irimia et al., 2014*). Consistently, heterozygous mice from a *Srrm4* mutant model, which exhibit similar levels of global microexon misregulation as human ASD individuals, displayed multiple hallmarks of ASD, including decreased habituation to stimuli and reduced preference for social interactions (*Quesnel-Vallières et al., 2016*). Moreover, a mouse model mimicking the ASD misregulation of a single microexon in *Cpeb4* also showed a range of autistic-like behavioral phenotypes (*Parras et al., 2018*). Therefore, we next set out to assess potential social defects in zebrafish mutant microexon lines. For this purpose, we implemented a custom version of the behavioral set-up developed in *Hinz and de Polavieja, 2017*; *Figure 5—figure supplement 1* and used *idtracker.ai* (*Romero-Ferrero et al., 2019*) to track activity and interactions of pairs of ~1-month-old juveniles (*Figure 5A*; see Materials and methods). For all lines except for the double *srrm3/4* mutant, which could not reach this age, we aimed at blindly assaying at least 10 pairs of each genotype combination (two heterozygous [Het-Het], two homozygous for the deletion [Del-Del] or one of each [Het-Del]), when possible. Of note, given the slower growth of *srrm3* homozygous mutants, we used WT-size-matched individuals, corresponding to ~3 months of age, to minimize potential technical and biological biases (see Materials and methods). For each fish on a pair, we quantified different parameters related to locomotion (speed, normal and tangential acceleration, distance traveled), anxiety (distance to the center of the arena) and leadership (time following the other fish in the pair). In addition, we quantified the distance between the two fish from a pair and the overall agreement between their movement vectors

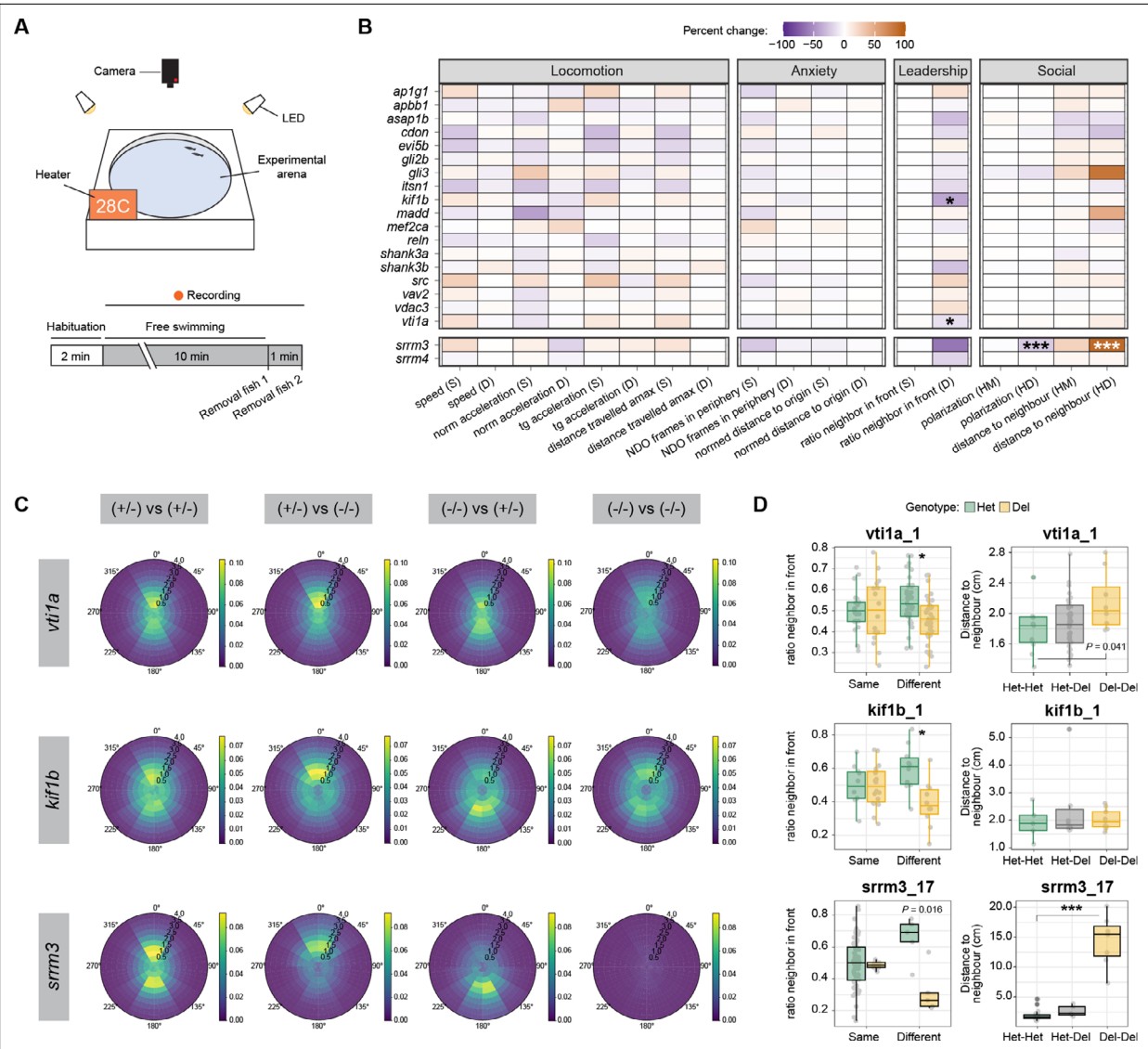

**Figure 5.** Impact of microexon misregulation on social behavior. (**A**) Schematic representation of the behavioral station (details in *Figure 5—figure supplement 1*) and experimental design for each tested pair of 30 dpf juveniles. (**B**) Heatmap showing the percent of change with respect to the control value for each main microexon and regulator deletion line for nine different parameters related to locomotion, anxiety or social behavior (full plots in *Figure 5—figure supplements 2–10*). For parameters referring to individual (locomotion, anxiety, and leadership), two comparisons are shown: the average of the fish from Del-Del pairs vs the control Het-Het pairs (homotypic pairs, same 'S'), or the value of the Del fish with respect to the Het one within each Het-Del pair (heterotypic pairs, different 'D'). For group parameters (polarization order and distance to neighbors), the following two comparisons are shown: Het-Het vs Het-Del (HM) and Het-Het vs Del-Del (HD). While 'ratio neighbor in front' (**D**) does not reach statistical significance under our cut-offs for *srrm3* mutants due to the small sample size (N=5; p=0.016), their strongly increased leadership can be observed in panels C and D. (**C**) Relative position map showing the position of the other fish of the pair (right genotype) with respect to the focal fish located at the center of the map (left genotype). For instance, (+/-) vs (-/-) shows the position of the Del fish respect to the focal Het one in a Het-Del pair. The merge plot of all fish pairs for *vti1a*, *kif1b*, and *srrm3* are shown. (**D**) Boxplots for the fish pairs shown in (**C**). Left: ratio in front of the non-focal fish. Lower values indicate higher leadership (i.e. more time in front). 'Same' corresponds to either Het-Het or Del-Del pairs and 'Different to Het-Del pairs, with values of individual fish plotted by genotype. Right: median of the distances between the two fish throughout the time course for each genotype pair combination. p-Values correspond to Wilcoxon Rank-Sum tests for the indicated comparisons. * 0.01<p ≤ 0.001, ** 0.001<p ≤ 0.0001, *** p<0.0001. Note that for all experiment, *srrm3* KO mutants (Del) were sized-matched (not aged-matched) with Het controls (see Materials and methods).

The online version of this article includes the following figure supplement(s) for figure 5:

**Figure supplement 1.** Custom behavioral station.

**Figure supplement 2.** Median speed by genotype for each main microexon and regulator line.

**Figure supplement 3.** Median absolute normal acceleration by genotype for each main microexon and regulator line.

*Figure 5 continued on next page*

*Figure 5 continued*

**Figure supplement 4.** Median absolute tangential acceleration by genotype for each main microexon and regulator line.

**Figure supplement 5.** Median distance traveled by genotype for each main microexon and regulator line.

**Figure supplement 6.** Time in the periphery by genotype for each main microexon and regulator line.

**Figure supplement 7.** Median normalized distance to origin by genotype for each main microexon and regulator line.

**Figure supplement 8.** Ratio neighbor in front by genotype for each main microexon and regulator line.

**Figure supplement 9.** Polarization by genotype pair combination for each main microexon and regulator line.

**Figure supplement 10.** Interindividual distance by genotype pair combination for each main microexon and regulator line.

(polarization), both of which give a measure of their social preference. Specifically, WT juvenile fish display a very strong social behavior, with small inter-individual distances and highly concordant movements (*Hinz and de Polavieja, 2017*).

We found no statistically significant differences (p<0.01) for any locomotion or anxiety related parameter for any microexon deletion or for either the *srrm3* or *srrm4* mutant lines (*Figure 5B*). This was the case when comparing the average values of the two homotypic pairs (Het-Het vs Del-Del; same 'S') or the values of Het and Del fish within heterotypic pairs (Het-Del; different 'D'), although some potential trends could be observed (*Figure 5B*, *Figure 5—figure supplements 2–10*). In contrast, *srrm3* mutants as well as two microexon deletions, *vti1a* and *kif1b*, showed significant defects in social parameters (*Figure 5B–D*). In particular, homozygous Del fish for the three mutations exhibited significantly higher leadership, that is an increase in the time spent in front of a paired Het fish, as visually evident in the heatmaps of relative positions (*Figure 5C*). Increased leadership suggests reduced social preference in the Del fish compared to the Het one, which tends to follow its pair closely. In line with this, the inter-individual distance was increased in Del-Del pairs compared to other genotype combinations for *srrm3* and *vti1a* (*Figure 5D*), while the polarization was decreased in *srrm3* mutant pairs (*Figure 5—figure supplement 9*). These differences in social behavior were relatively minor for *vti1a* and *kif1b* mutants, but very strong for *srrm3* fish, which displayed a dramatically reduced social behavior. Importantly, this was not due to the difference in age between the heterotypic pairs, as it could be also observed in recordings of groups of five homozygous mutants compared to groups of WT ones, all of 5 months of age (*Video 2* and *Video 3*). The altered social behavior and increased leadership are likely caused by the visual impairment of *srrm3* homozygous mutants (*Ciampi et al., 2022*).

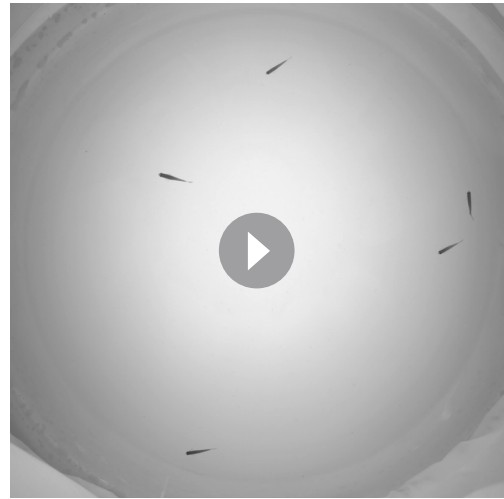

**Video 2.** Group of five srrm3 mutant juvenile fish. Also accessible at https://data.mendeley.com/datasets/3b3zx4cfg9/1.
https://elifesciences.org/articles/104275/figures#video2

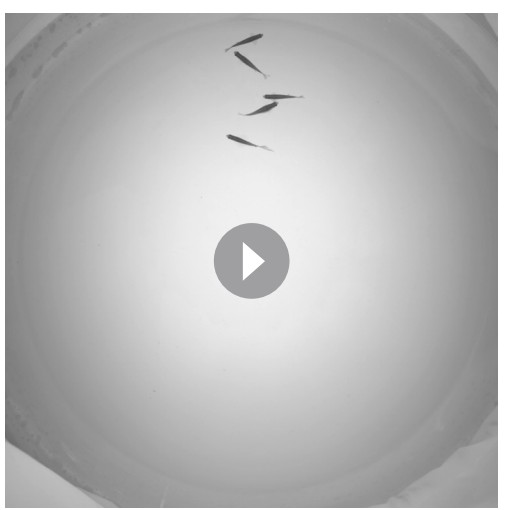

**Video 3.** Group of five WT juvenile fish. Also accessible at https://data.mendeley.com/datasets/3b3zx4cfg9/1.
https://elifesciences.org/articles/104275/figures#video3

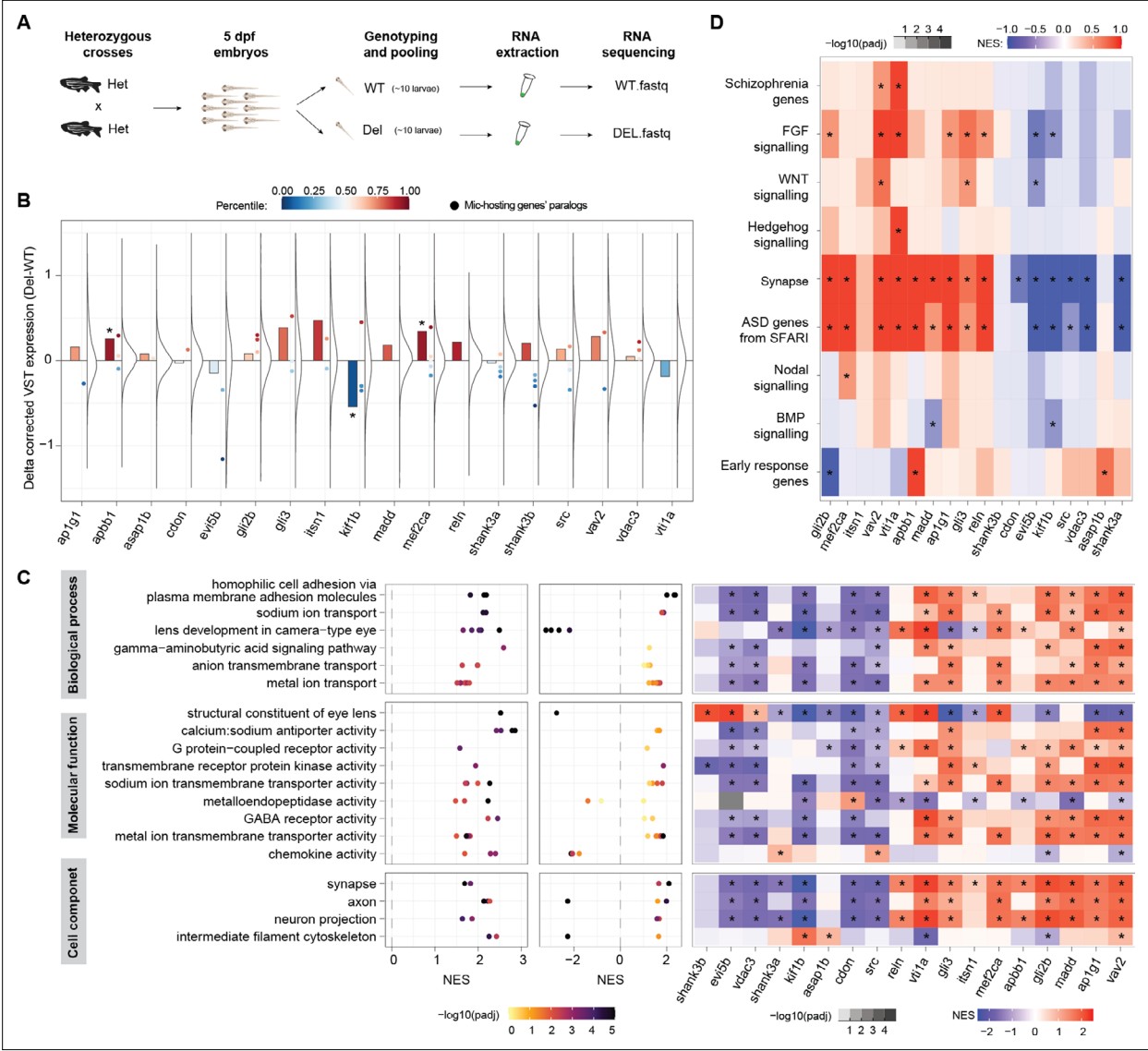

**Figure 6.** Transcriptomic analyses of 5 dpf larvae suggest potential compensatory changes. (**A**) Schematic representation of the experimental design. (**B**) Distribution of changes in gene expression (Δ corrected variance stabilizing transformation [VST] expression) between the Del and WT larvae for each main microexon deletion line, as well as change of the host gene (barplots) and of closely related paralogs (chordate or younger origin according to Biomart) (dots; Mic-hosting genes' paralogs). The color scale indicates the percentile of the expression change within the overall distribution. Asterisk on bars indicates change in the bottom or top decile. (**C**) Left: Gene Ontology (GO) categories (dots) clustered in related groups (rows) that are enriched among genes globally changing upon individual microexon deletions. Middle: Normalized Enrichment Scores (NES; X-axis) and adjusted p-values (color code) for the same GO categories in the comparison of *srrm3* mutant larvae and WT siblings. Right: heatmap showing the NES for the gene sets comprising the union of all GO categories within each group for each specific microexon comparison. Stars indicate adjusted p-value <0.01. (**D**) NES for specific gene sets (***Supplementary file 7***) with relevance in neurobiology and/or development for each microexon deletion comparison.

The online version of this article includes the following figure supplement(s) for figure 6:

**Figure supplement 1.** Batch correction of RNA-seq data.

**Figure supplement 2.** Enriched GO categories.

## Transcriptomic analyses suggest potential compensatory changes through regulation of neural-related pathways

Given the minor phenotypes we observed for most microexon deletions, it is conceivable that defects in individual protein functions are compensated for by other molecular changes. Hence, we aimed at assessing global transcriptomic alterations in microexon mutant lines. For this purpose, we generated RNA-seq data for pools of ~10 5 dpf homozygous mutant larvae (Del) and wild-type siblings (WT)

for the 18 microexon deletions, as well as for the *srrm3* mutant line (*Figure 6A*, *Figure 6—figure supplement 1* and *Supplementary file 6*; see Materials and methods). As expected, all microexon deletions led to minor changes in expression for most genes (absolute delta corrected variance stabilizing transformation [VST] expression <0.5; *Figure 6B*). To identify potential compensatory changes, we first focused on changes in the expression of the host gene as well as its closely related paralogs. A few microexon deletions resulted in substantial relative changes in the expression of the host gene (e.g. *apbb1*, *mef2ca*, *kif1b*), particularly towards increased levels, and/or of some of the paralogs (e.g. *apbb1*, *evi5b*, *gli3*, *mef2ca*, *shank3b*; *Figure 6B*). Some of these changes may have straightforward interpretation. For instance, inclusion of the 24-nt microexon in the human ortholog of *mef2ca* has been reported to increase its transactivation activity *Zhu et al., 2005*; therefore, an increase in overall expression levels of the isoform without the microexon could conceivably compensate for the deletion of the microexon. Next, we reasoned that, while individual genes may not exhibit strong expression changes, coordinated changes in specific cellular pathways could identify significant patterns shared across multiple microexon deletions. For this purpose, we summed the absolute changes in expression (WT vs Del) for each gene across the 18 microexon deletions and performed a positive Gene Set Enrichment Analysis (GSEA) with these joint differential expression values. This revealed various groups of related GO categories that showed coordinated gene expression changes across the microexon lines (*Figure 6C*). Remarkably, these categories were highly related to neuronal function and organization (e.g. channels, G-protein-coupled receptor activity, synapse, axon, etc.). Many of these categories were also differently expressed in *srrm3* mutant larvae (*Figure 6C*). Importantly, the same coordinated changes were not observed in all microexon deletions, or they did not occur in the same direction (up- or down-regulation; *Figure 6C*). This is in line with the patterns observed for neurite growth (e.g. *vav2* and *evi5b*; *Figure 3*) or for other phenotypic traits (e.g. insulin secretion in beta cells *Juan-Mateu et al., 2023*), and it suggests that different microexons within the *srrm3*-regulated program may have opposite impacts. Finally, we looked at expression changes in manually curated gene sets of special relevance for neurobiology and/or embryo development, including gene orthologs associated with ASD or schizophrenia, as well as multiple signaling pathways. Some of these sets were widely and significantly misregulated across microexon lines, particularly those related to ASD and the synapse (*Figure 6D*). In summary, while microexon deletions lead to mild changes in expression of individual genes, they often result in significant, coordinated effects on specific neurally related pathways, which may suggest molecular compensation.

## Discussion

Despite the major defects observed for the loss-of-function models of their master regulators (*Calarco et al., 2009*; *Nakano et al., 2012*; *Quesnel-Vallières et al., 2015*; *Nakano et al., 2019*; *Torres-Méndez et al., 2022*), it is still unknown how individual microexons underlie those phenotypes. In this study, we set out to address this question by generating and characterizing 18 microexon deletion zebrafish lines, along with mutant lines of *srrm3* and *srrm4*. To phenotypically characterize these lines, we implemented a battery of tests, aiming at evaluating a wide range of potential neural defects at multiple levels: cellular (neuritogenesis), tissue (gross morphology and axon guidance), behavioral (individual stimuli response, collective interactions), and molecular (transcriptome-wide gene expression patterns). These assays provide a clear picture: while loss of function of the master regulators had strong defects at nearly all levels, particularly for *srrm3* (consistent with *Nakano et al., 2019*; *Ciampi et al., 2022*), all individual microexon deletions were viable and exhibited mild or no detectable alterations in nearly every feature we measured. These results are also in line with those obtained by *Calhoun et al., 2024*, who assessed larval brain activity, morphology, and behavior for 45 microexon mutants (only two in common with our study).

These results thus raise a major fundamental question: why do individual microexon deletions lead to such mild phenotypes? There are several potential explanations, most of which are not mutually exclusive. First, the easiest explanation is that most microexons could simply lack any relevant function. However, this is unlikely for the majority of the microexons we have studied, as they are conserved from zebrafish to human, for at least 450 million years of evolution. Moreover, functions at the protein level have been reported for the mammalian orthologs of a few of these microexons, including the modulation of protein-protein interactions (*apbb1 Irimia et al., 2014*, *itsn1 Dergai et al., 2010*, *src Dergai et al., 2010*), transactivation activity (*mef2ca Zhu et al., 2005*) or motor

activity (*kif1b Matsushita et al., 2009*). In addition, we uncovered here a difference in GEF activity between the two isoforms of *vav2*, which is in accordance with the impact of the microexon deletion on neurite growth (*Moon and Gomez, 2010*).

Second, another set of trivial explanations relates to the experimental design and includes that (i) we may not be assessing the traits that these microexons are impacting (e.g. complex neuronal morphologies, circuit function, other behavioral patterns, specific neuronal types, etc.), (ii) we may not have the sensitivity to robustly measure the magnitude of the changes caused by microexon removal, and/or (iii) the effects may not manifest under optimal laboratory conditions. The latter has been shown to be key, for instance, for embryo growth phenotypes upon *Mbnl3* KO in mouse, which only manifested when the mothers were subjected to calorie-restricted diets (*Spruce et al., 2022*), or for shadow enhancers in fruit fly, whose effects were only appreciated upon suboptimal temperature conditions (*Frankel et al., 2010*).

Third, related to the second point, it is also possible that relatively small perturbations can be compensated for by other molecular changes, particularly under optimal environmental conditions. While this is a very complex hypothesis to test, we have uncovered substantial changes in gene expression in the host gene and/or related paralogs (a classic compensatory mode *El-Brolosy et al., 2019*), as well as significant coordinated expression changes in gene categories enriched for neural functions. In this scenario, the mild effects of single microexon deletions could be neutralized. In contrast, the global microexon misregulation upon *srrm3/4* mutation would introduce a high number of small changes impacting multiple pathways that could not be properly compensated for altogether, thus causing large phenotypes. Interestingly, it is possible that, despite its many positive aspects as a model organism, zebrafish could be a less suited system to assess the mild phenotypic effects expected for microexon deletions, given the additional whole genome duplication of teleosts compared to most other vertebrates (*Amores et al., 1998*). This higher number of paralogs confers zebrafish much higher genetic redundancy compared to mammals, which can be used for potential compensatory changes. Consistent with this idea, various studies have identified robust phenotypes for individual microexon deletions in vivo in mouse models (e.g. *Wang et al., 2015*; *Gonatopoulos-Pournatzis et al., 2020*; *Poliński et al., 2025a*; *Poliński et al., 2025b*), although publication biases cannot be excluded (i.e. towards positive results, e.g. see *Matalkah et al., 2022*, and due to the higher number of mouse vs zebrafish studies).

Finally, relatedly, another potential explanation may reside in the very nature of the molecular role of microexons. Given that they encode for only a few amino acids and they exhibit dramatic cell type specificity, microexon inclusion is expected to contribute to the specialization of a protein for a given context (e.g. the synapse) with respect to the broadly expressed microexon-skipping isoform. In this scenario, both protein isoforms would be partly redundant by definition, as they play equivalent roles but specialized for different contexts, much like two gene paralogs (*Mantica and Irimia, 2025*). Therefore, and particularly under optimal environmental conditions, this redundancy is likely to mask the effects of the individual microexon deletions.

Nevertheless, whatever the explanation(s) behind the lack of strong phenotypes observed for most microexon deletions, the exceptional conservation of these exons for hundreds of millions of years can be considered an evolutionary paradox. This case is reminiscent of other highly conserved genomic elements, whose deletions usually lead to viable animals, with little or no alterations, such as the ultraconserved elements (*Ahituv et al., 2007*) or some long non-coding RNAs (*Goudarzi et al., 2019*). Further research should solve this paradox by providing the missing pieces (i.e. lack of sensitivity or adequate conditions, compensation, redundancy), which will help to better understand the roles and physiological relevance of individual microexons and, more generally, of tissue-specific alternative exon programs.

## Materials and methods
### Definition of neural alternative exons and microexons
Inclusion tables for zebrafish samples were downloaded from *VastDB* (*Tapial et al., 2017*; vastdb. crg.eu). To define neural-enriched exon and microexon programs, referred to simply as neural exons and microexons, we parsed these RNA-seq datasets to identify tissue-enriched exons using the script *Get_Tissue_Specific_AS.pl* script (*Martín et al., 2021*; https://github.com/vastdb-pastdb/pastdb),

requiring the following cut-offs: (i) absolute difference in the average exon inclusion level (using the percent-spliced-in metric, PSI) between the target tissue and the averages across other tissues of |ΔPSI|>15 (`--min_dPSI 15`), (ii) global |ΔPSI|>25, that is difference between target tissue inclusion average and the average of all other tissues combined (`--min_dPSI_glob 25`), (iii) a valid average PSI value in at least N=5 tissues (`--N_groups 5`), and (iv) sufficient read coverage in at least n=1 samples per valid tissue group (`--min_rep 1`), that is score VLOW or higher as provided by *vast-tools* (*Tapial et al., 2017*). A config file with the tissue groups for each RNA-seq sample was provided as input (*Supplementary file 5*). To better capture all microexons with biased inclusion in certain tissue groups, we excluded from the calculations for specific tissue groups the tissues with known or expected partial overlap of microexon inclusion (e.g. neural, endocrine pancreas, muscle, and heart), as provided in the column EXCLUDED in the config files. All neural microexons in zebrafish are provided in *Supplementary file 1*. Human neural microexons were obtained from *Juan-Mateu et al., 2023*, which were defined with similar parameters, but requiring sufficient read coverage in at least two samples per valid tissue group (`--min_rep 2`), whereas we only required one.

## Functional, evolutionary, and regulatory characterization of zebrafish microexons

To visualize the inclusion levels of neural microexons during zebrafish development, inclusion patterns of all microexons across developmental stages (from egg to 7 dpf) were collected from VastDB. GO enrichment analysis for genes harboring zebrafish microexons was performed using the software ClueGO (*Bindea et al., 2009*). The gene background set (N=15,811) consisted of genes containing exons of any kind with the expression level requirements used to define tissue-enriched exons (see above). In particular, they were derived by running the *Get_Tissue_Specific_AS.pl* script with the same parameters used for the tissue-enriched microexon call and requiring sufficient read coverage in neural samples. Exon conservation between zebrafish and human (and vice versa) was assessed as follows. First, we generated gene orthogroups between zebrafish and human by complementing the gene orthologies from *Ensembl* (v80) with those provided in *Ciampi et al., 2022*. We then ran *ExOrthist main* (v1.0.2; *Márquez et al., 2021*) using the default conservation cutoffs for long evolutionary distances to infer exon orthogroups. In order to increase the sensitivity of the exon orthology calls, we also considered non-annotated exons identified by *vast-tools* (*Tapial et al., 2017*; `--extraexons option`), and we provided pre-computed *liftOver* hits between the two species (computed with the *ExOrthist* companion script *get_liftovers.pl*) to be directly integrated in the exon orthogroups (`--bonafide_pairs option`). Genome conservation was estimated based on the resulting exon orthogroups, while neural-enriched regulatory conservation was assessed based on the ΔPSI values (average PSI in neural samples - average PSI in other tissues) from the *Get_Tissue_Specific_AS.pl* script as described above (using the `--test_tis` option). If the exon had sufficient read coverage, it was considered regulatory conserved if ΔPSI ≥15. To evaluate regulation by *srrm3/4* in zebrafish, we obtained the ΔPSI for each exon with sufficient coverage between mutant and control samples for three different pairs of samples: (i) retina from *srrm3* KO or WT 5 dpf larvae (*Ciampi et al., 2022*), (ii) retina from *srrm3/4* double KO and *srrm4* KO 5 dpf larvae (*Ciampi et al., 2022*), (iii) whole *srrm3/4* double KO and WT 5 dpf larvae (this study, see 'Generation of RNA sequencing and transcriptomic analysis'). We then selected the lowest ΔPSI value for each exon and plotted their distributions for each exon type (*Figure 1E and F*). Human overexpression RNA-seq data in HEK293 cells and a matched control was obtained from *Torres-Méndez et al., 2019*. In brief, the experimental condition corresponded to a Doxycycline-induced overexpression of the most active enhancer of microexons domain (eMIC)-containing protein tested in *Torres-Méndez et al., 2019*, obtained from the amphioxus Srrm234 pro-ortholog. The control corresponded to GFP overexpression.

## Zebrafish husbandry and genotyping

Fish procedures were approved by the Institutional Animal Care and Use Ethic Committee (PRBB–IACUEC). Zebrafish (*Danio rerio*) were grown in the PRBB facility at 28 °C 14 hr light/10 hr dark cycle. All lines generated were reared under standard facility conditions, with the exception of the *srrm3* line, which needed special conditions. These special conditions consisted of water with a salinity of 5 ppm (parts per million) and no renewal of water for 48 hr, which allowed higher survival of rotifers (*Brachionus plicatilis*) and thus kept higher food availability for longer periods of time. This allowed

homozygous *srrm3* mutants to feed and survive beyond 12–20 dpf, when they would otherwise die due to issues derived from their impaired vision (*Ciampi et al., 2022*). These homozygous *srrm3* mutants grew more slowly and were substantially smaller than their siblings. Therefore, we separated them at ~1 month post-fertilization (mpf) and reared them separately to avoid competition with their siblings and to be able to closely monitor them.

All zebrafish mutant lines were generated using the Tg(HuC:GFP) line (*Park et al., 2000*) as background, kindly provided by Elisa Martí's laboratory at the Institute of Molecular Biology of Barcelona (IBMB). To preserve colony health and maintain genetic diversity, an AB wild-type line was regularly incorporated. Fish were outcrossed with AB individuals every two generations to prevent inbreeding, and HuC:GFP fluorescent embryos were manually selected. For *Figure 2E*, fish size was measured by determining the standard body length, following the methodology described by *Näslund, 2014*. For adult fish genotyping or 3–5 dpf larvae, a piece of the caudal fin was cut and disaggregated using 100 μL of NaOH 50 mM for 15 min at 96 °C. To neutralize the reaction, 10 μL of 1 M Tris-HCl pH = 7.4 was added. 2 μL were used as templates for PCR with the pairs of primers designed to amplify the genomic region of interest (*Supplementary file 4*) and using GoTaq Flexi DNA Polymerase kit (Promega, M7806). PCR products were run on agarose (Invitrogen, 16500500) gels at 2% in TBE. Tissue samples used for microexon pattern validations in *Figure 1—figure supplement 1B* were obtained from dissections of adult Tg(HuC:GFP) WT fish and RNA extracted using RNeasy Mini Kit (QIAGEN, 74134), following the manufacturer's instructions.

Individual and massive crosses were carried out in crossing tanks and the resulting eggs were screened for overall health from embryo collection to 5 dpf. Embryos were collected into Petri dishes with E3 medium with methylene blue and placed into the incubator at 28 °C. For each experiment, we used either homozygous (Del) or heterozygous (Het) mutant incrosses, as specified. In the case of the double *srrm3/4* mutant, for which adults were not viable, and given that *srrm4* homozygous mutants displayed largely no phenotypes, we used *srrm3+/+,srrm4-/-* or *srrm3+/-,srrm4-/-* as controls, to which we refer as W/D and H/D in *Figures 3 and 4*, respectively, and WT (*srrm3+/+,srrm4-/-*) or Het (*srrm3+/-,srrm4-/-*) in figure supplements 3-13 for simplicity. Mutant lines of the regulators were generated in *Ciampi et al., 2022*.

## Generation of microexon deletion lines using CRISPR-Cas9

We selected pairs of guide RNAs (gRNAs) to target the 21 selected microexons using CRISPRscan (https://www.crisprscan.org/; *Moreno-Mateos et al., 2015*). The pairs of gRNAs were chosen to be positioned one upstream and one downstream of the microexon region, spanning a maximum of 100 bp from the edges of the microexon. This design aimed to minimize any potential impact on the surrounding intronic regions. Sequences for all gRNAs are provided in *Supplementary file 3*. The gRNA synthesis protocol was adapted from *Gagnon et al., 2014* and optimized for efficient in vitro transcription using MEGAshortscript T7 (Thermo Fisher, AM1354). This protocol allows for the transcription of up to 96 sgRNAs in a single well plate. Briefly, the protocol involved annealing oligos to create the template for T7 gRNA transcription, followed by template purification using QIAquick (QIAGEN, 28104) and quantification with Qubit dsDNA BR kit (Thermo Q32850). gRNA was synthesized using the MEGAshortscript T7 (Thermo, AM1354), purified by precipitation, and finally quantified using the Qubit RNA HS assay (Thermo, Q32852). The final gRNAs were then ready for use for CRISPR-Cas9 injections. Embryos were injected at the 1–2 cell stage with either single or double gRNAs. Initial single gRNA injections were performed to evaluate the efficiency of individual gRNAs and to enhance the effectiveness of subsequent double gRNA injections for establishing the stable deletion lines. The injection mix contained Cas9 protein (PNABio, CP01-50) at 300 ng/μL, sgRNAs at 50 ng/μL, and Phenol Red solution (Merck, P0290) in sterile Milli-Q water.

gRNA efficiency was validated using a pool of 16 embryos at 48 hours post-injection (hpi) for each injection. DNA extraction was performed using a standard HotSHOT protocol (*Samarut et al., 2016*). Fluorescent PCR was performed on genomic DNA, followed by analysis with a Fragment Analysis Service (GeneScan, Thermo) to achieve single-nucleotide resolution of mutations induced by the sgRNAs. GeneScan 500 color ROX (Thermo, 4310361) was used as a reference size marker. The most effective sgRNAs were selected based on their capacity to produce multiple deletions near the microexon region. Double gRNA validation was performed using standard semi-quantitative PCR.

## Validation of microexon inclusion levels in WT and mutant lines

Primers for RT-PCR assays to validate microexon inclusions were taken from VastDB (*Tapial et al., 2017*) or designed to span the upstream and downstream exon, optimizing the detection of the two bands for inclusion and exclusion isoforms while avoiding excessive amplification bias of the short isoform (*Rukov et al., 2007*). Primer sequences are provided in *Supplementary file 4*. For RNA extraction, embryos or fresh tissues from embryos and adults were homogenized in TRIzol (Thermo, 15696018) using a MiniBeadBeater (Biospec Products) for 40 s, along with the addition of glass beads (Merck, G8772-100G). RNA was extracted to validate the inclusion patterns of exons in WT tissues (*Figure 1—figure supplement 1B*) and in the knockout (KO) lines (*Figure 2—figure supplement 1*) using TRIzol/chloroform extraction protocol. Final RNA concentrations were quantified using a Qubit Fluorometer with the RNA HS Assay Kit (Invitrogen, Q32855). Total RNA was then reverse transcribed to synthesize cDNA using SuperScript III Reverse Transcriptase (Invitrogen, 18080044), according to the manufacturer's guidelines. cDNA was quantified using Nanodrop and 2 µL of cDNA at 100 ng/µL was used for validation, employing standard PCR conditions with variations similar to those used for genotyping; initial denaturation at 96 °C for 3 min, followed by 40 cycles of 20 s at 96 °C, 30 s of annealing at 56 °C, and a 1 min elongation at 72 °C, concluding with a final elongation step of 5 min at 72 °C. The resulting PCR products were separated by gel electrophoresis on a 2.5% to 4% ultrapure agarose gel (Invitrogen, 16500500) on SB buffer containing SYBR Safe DNA Gel Stain (Invitrogen, S33102), depending on the size of the products.

## Immunofluorescence of Mauthner cells (3A10)

For whole-mount immunostaining of Mauthner cells, incrosses of heterozygous parents from micro-exon and regulator mutant lines were used. Embryos were initially treated with 0.003% 1-phenyl 2-thiourea (PTU, Sigma-P7629) starting at 20 hpf to inhibit pigment development. At 48 hpf, the embryos were dechorionated and fixed overnight in 4% paraformaldehyde (PFA) at 4 °C. The immunostaining protocol began with washing the embryos in 0.2% Triton X-100 in phosphate-buffered saline (PBSTx), followed by a permeabilization step using cold acetone at –20 °C for 10 min. The samples were then washed again in PBSTx and blocked for 1 hr in a blocking solution consisting of PBSTx and 10% goat serum. Subsequently, the embryos were incubated at 4 °C for 24 hr with the primary antibody anti-3A10 (AB_531874, Developmental Studies Hybridoma Bank), a neurofilament marker. After this incubation, the embryos were thoroughly washed in PBSTx before being incubated with the secondary antibody, Alexa goat anti-mouse 594 (1:500, Thermo Fisher Scientific, A-11034). Embryos were then genotyped as described above and dorsally mounted in low-melting agarose. Imaging was performed using a standard fluorescent microscope (Leica DMI 6000 B), with images captured using a 20 x objective. The evaluation of neural patterning and axon guidance defects was conducted using Fiji-ImageJ, focusing on the observation and quantification of the presence or absence of the two Mauthner cells and their midline crossing. A total of 15–70 embryos were analyzed for each mutant line from the main founders.

## Generation of neuronal primary cultures and quantification of neurite outgrowth

We developed a protocol for isolation, culture, and quantification of neurite outgrowth from larval zebrafish neurons. As a general experimental design, we first performed an initial screen, where we used homozygous mutant (Del) incrosses from each main founder, with independent incrosses of Tg(*HuC:GFP*) used as matched control samples for each batch. For those microexons for which significant differences were obtained, we performed additional biological replicates of the main founder and tested the second founder as well. Finally, for the most promising cases (*evi5b* and *vav2*), we generated heterozygous incrosses and grouped sibling larvae by genotype before cell isolation. To assay *srrm3*, *srrm4,* and double *srrm3/4* mutants, we followed the same strategy, given that srrm3 and srrm3/4 homozygous mutants were not viable.

### Fluorescent activated cell sorting (FACS) of zebrafish neurons

Embryos intended for neuronal cell culture were bleached between 24 and 30 hpf, prior to hatching, to prevent contamination. The bleaching protocol consisted of immersing the embryos in a 0.1% sodium hypochlorite solution for 5 min, followed by neutralization in 0.5 µg/L thiosulfate for an

additional 5 min. Finally, the embryos were transferred to E3 medium containing methylene blue and stored at 28 °C. At 48 hpf, the embryos were dechorionated using Pronase (Sigma, 10165921001) at a final concentration of 1 mg/mL in E3 medium. Embryos were gently swirled until the chorion softened, usually within 3–5 minutes. They were then transferred to a Petri dish with fresh E3 medium supplemented with 1% penicillin-streptomycin (PNS) to remove any residual Pronase and prevent tissue degradation. Prior to dissociation, embryos were anesthetized by rapid cooling, dechorionated with pronase, and organized into sample groups of 50 embryos per tube, with two biological replicates per condition (control and mutant).

Dissociation was performed under strict sterile conditions. Embryos were immersed in 400 µl of Neurobasal medium (NB) and 100 µl of 0.5% trypsin-EDTA. Three rounds of enzymatic and manual dissociation, each lasting 10 min, were performed using sterile pestles. After dissociation, samples were centrifuged for 3 min at 5000 revolutions per minute (rpm). The supernatant was removed, and fresh NB medium with supplements (1% P/S, 1% N2, 1% Fungizone, 2% B27, 1x L-glutamine, and 2% fetal bovine serum) was added to stop the trypsinization reaction. The cell suspension was then filtered through a 30-µm cell strainer and kept on ice until sorting. Before sorting, samples were stained with propidium iodide (PI, 1:1000) to assess viability, and then filtered through a 20-µm filter. A high-resolution cytometer (BD Influx Cell Sorter) was used to detect Forward scatter (FSC), side scatter (SSC), and fluorescence detection lasers for FITC (488 nm), PE-TR-PI were used to gate for live, single, GFP+ cells. Finally, GFP+ cells (ranging from 23–49% of the live population) were sorted and collected in pools of 200,000 cells in 300 µL of NB inside sterile Eppendorf tubes for various downstream applications, including RNA extraction and neuronal cell culture. All data analysis was performed using FlowJo software.

## Primary cultures of sorted zebrafish neurons

The culture protocol was developed to be able to obtain a zebrafish neuronal culture from the mutant lines. The method was based on previous studies on culturing zebrafish cell types (*Sakowski et al., 2012*; *Sassen et al., 2017*). The starting material for the culture development was the isolated neurons obtained through FACS, which were centrifuged for 5 min at 5000 rpm. Pellet cells were resuspended in 300 µL of Neurobasal (NB, Thermo, 21103–049) media with supplements (1% Penicillin streptomycin- 1% N2-, 1% Fungizone-, 2% B-27, 1 x L-Glutamine-) and plated onto 8-well glass-bottom slides (IBIDI: 80827) coated with 500 µg/mL Poly-D-Lysine (coated for 3–4 hr at 37 °C). Cells were incubated at 28 °C with 5% $CO_2$. Cells obtained were fixed 15 min in 4% Paraformaldehyde (PFA) at different time points (4 hr, 8 hr, 10 hr, 24 hr, 48 hr, 72 hr). Then cells were washed in 1xPBS and stored in 1xPBS supplemented with 0.02% Sodium Azide (Sigma, S2002) to avoid contaminations, for posterior immunostaining and neurite growth studies. Cultured neurons exhibited robust progression in neuronal development and growth (*Figure 3—figure supplement 1*), enabling comprehensive study and quantification of neurite length and potential alterations across various genetic backgrounds.

## Immunofluorescence of cultured neurons

For the main quantification assays, cells were fixed at 10 and 24 hr after plating (hap) with 4% paraformaldehyde (PFA) for 15 min. After fixation, cells were washed three times with 1 x PBS and stored at 4 °C in 300 µL of 1 x PBS containing 0.02% sodium azide to prevent contamination. For immunofluorescence, cells were permeabilized three times for 5 min with 0.1% Triton-X100 in 1 x PBS with 5% normal goat serum (NGS), followed by a 1 hr block at room temperature (RT) in 0.1% Triton-X100 in 1 x PBS with 1% NGS. Primary antibodies were incubated overnight at 4 °C in a humidified chamber: mouse anti-acetylated tubulin (1:400, BioLegend, 802001) and rabbit anti-GFP (1:500, Millipore, AB3080P). After incubation, samples were washed three times for 10 min in 0.1% Triton-X100 in 1 x PBS with 1% NGS, followed by a 2 hr incubation at RT with secondary antibodies: Alexa Fluor 594 goat anti-mouse (1:500, Thermo Fisher, A-11005) and Alexa Fluor 488 goat anti-rabbit (1:500, Thermo Fisher, A-11034). Finally, samples were washed three times with 1 x PBS and stained for 1 hr with Hoechst (1:1000, Sigma, B2261) in sterile water for nuclear staining. After staining, samples were stored in 1 x PBS with 0.02% sodium azide at 4 °C until imaging.

## Imaging and analysis neurite length

We used a Leica TCS SP8 inverted confocal microscope to image neuronal culture samples. Neuronal cultures were performed in 8-well chamber slides to enhance imaging efficiency and maintain consistent conditions across all samples. Images were automatically acquired for all wells. For each well, we captured images at 5 randomly selected positions, covering the entire growth surface (1 cm²). Each position was imaged with 15 Z-stacks, with a stack depth of 1 μm per slice. We employed three lasers for imaging: 405 nm for DAPI (nuclei), 488 nm for HuC:GFP, and 594 nm for acetylated tubulin (Ac. Tub). The acquisition parameters included: an image size of 512x512 pixels, a scanning speed of 600, bidirectional phase-X correction, a zoom factor of 1.5 x, and images were captured using a ×20 magnification objective. For each experimental line, we analyzed at least two different wells, with five positions imaged per well. Neurite length was manually quantified using ImageJ, measuring from the tip of the projection to the soma. A total of 100 measurements of neurite length were taken for each well. Note that, while identification of entire individual neurites becomes complicated at late time points, at the time points used in this study, 10 hap and 24 hap, most neurites could be individualized and unequivocally associated to a soma. Dubious cases were not measured and used for the analyses. Statistical differences in the main lines of deleted microexon and regulator were assessed by performing 10,000 bootstrap resampling. This involved randomly selecting 50 data points per experiment per genotype and conducting a Wilcoxon test between control and mutant neurons, or between siblings of different genotypes (*Figure 3—figure supplement 2*). When more than one founder was tested, an ANOVA was used to assess differences between genotypes.

## SRF promoter activation assay upon *vav2* overexpression in COS cells

To determine VAV2 protein activity, we used a luciferase reporter assay based on SRF promoter activation. In brief, $1\times10^6$ exponentially growing COS cells were transfected with 1 μg of GFP-VAV2_MIC- (pAA7) or GFP-VAV2_MIC+ (pMCL81) and the firefly luciferase reporter pSRE–Luc (1 μg; Addgene) and the vector constitutively expressing the Renilla luciferase pRL–SV40 (3 ng) (Addgene), using Lipofectamine 2000 (Cat. #11668019, Invitrogen). After 48 hours in complete medium, cells were washed in serum-free media and lysed in Passive Lysis Buffer (Cat. #E1960; Promega). The luciferase activity obtained in each condition was recorded using the Dual–Luciferase Reporter System (Cat.#E1960; Promega) according to the supplier's recommendations in a Lumat LB 9507 luminometer (Berthold). To generate the expression vector encoding EGFP-Vav2Onc (pNM115 *Lorenzo-Martín et al., 2020*), EGFP alone, or full-length zebrafish *vav2* cDNA with or without microexon, the plasmid pKES19 (*Schuebel et al., 1998*) was digested with BstXI, filled-in, and cloned into the SmaI-linearized pEGFP-C2 vector (Clontech-Takara Bio, Catalog No. 632481). The raw values obtained were normalized according to the activity of the Renilla luciferase recorded in each sample. Final values are represented as the fold change of the normalized luciferase activity obtained when compared to the GFP-VAV2_MIC- sample. In all cases, the abundance of the ectopically expressed proteins under each experimental condition was verified by analyzing aliquots of the cell extracts used in the luciferase experiments by immunoblot. Antibodies: anti-GFP (Cat. No. MMS-118P, Covance, Princeton, NJ, USA; 1:2,000 dilution), anti-β-actin (Cat. No. sc-47778, Santa Cruz; 1:1,000 dilution).

## NanoBRET protein-protein assay between EVI5B and RAB11

Full-length zebrafish EVI5B with or without the microexon was cloned into pHTNW and RAB11A into pNLF1W vectors *Yang et al., 2018* by Gateway LR reaction (Thermo Fisher Scientific) following the manufacturer's indications. HEK293T cells were plated in 24-well plates at a density of $1.4\times10^5$ cells per well. Cells were transfected with 500 ng of pHTNW-EVI5B, 5 ng of pNLF1W-RAB11A, 0.75 μL of Lipofectamine 3000 and 1 μL of P3000 Reagent (Thermo Fisher Scientific). After 20 hr, each transformation was re-plated in four wells of 96-well plates at a density of $1\times10^4$ cells per well for duplicate control and experimental samples (technical replicates), and protein-protein interactions were analyzed with NanoBRET Nano-Glo Detection System kit (Promega) following manufacturer's instructions. Each transformation experiment was performed at least twice. The corrected NanoBRET ratio was calculated according to the manufacturer's instructions and corrected as previously described (*Yang et al., 2018*). NanoBRET signal for EVI5B with or without the microexon co-transfected with or without RAB11A showed strong interaction signal, but similar for both EVI5B isoforms (*Figure 3I*).

## Evaluation of larval activity and response to stimuli using the DanioVision system

### Experimental design and procedures

DanioVision experiments were conducted using 5 dpf embryos placed in 48-well plates filled with E3 medium. The experiments were performed blindly, utilizing embryos from heterozygous incrosses of each microexon mutant line, as well as the regulator mutant lines. Animals were randomly assigned to the 48-well plate 1 hr prior to the experimental start to let them habituate to the environment. The experimental protocol in the Danio Vision chamber involved an initial 30 min of acclimatization, which included 5 min of habituation followed by 25 min of baseline activity recording. This was followed by consecutive rounds of dark-light transitions, consisting of three dark and two light phases. Finally, we assessed the startle response of the animals by delivering 30 mechanical tappings at one-second intervals. Video recordings were done at 30 frames per second with a sensitivity detection of 110. For pixel-based measures (activity [% delta pixels]), the moving pixel threshold was adapted to lighting and background (11–16 pixels). After completing the experimental procedures, the animals were collected and genotyped. Dead or unhealthy larvae were excluded from the analysis.

### Data export, processing, and analysis

Using the EthoVisionXT version 17.0 software, the mean activity data (% delta pixel) was exported in 1 s intervals. For analysis purposes, technical replicates collected from the same clutch on the same day were combined. Larvae with unclear genotypes were excluded from the downstream analysis. For the computation of activity and visual stimulus transition parameters, activity values were averaged by minute. Noisy trials with low video quality or inconsistent stimulus responses of the WT larvae were excluded prior to the analysis. The mean activity (% delta pixel) was subsequently calculated per larva for the baseline (B) period (experimental time 5–30 min), the dark (D) periods (experimental time 30–40 min, 50–60 min, 70–80 min) and the light (L) periods (experimental time 40–50 min, 60–70 min). For the light-dark and dark-light transitions, the difference in activity (% delta pixel) 1 min before and 1 min after the stimulus was obtained for each transition and averaged across same-type transitions. Differences were computed as follows: light to dark (LD)=activity 1st min dark - activity last min light; dark to light (DL)=activity 1st min light - activity last min dark. For the tapping parameters, larvae not responding to any of the first 3 taps were excluded from boxplots and statistical analyses as low responders (on average 1.3 larvae per genotype across experiments). The mean activity for each larva at tap number 1 (Act1 tap), 3–5 and 21–30 was obtained. To measure a drop in responsiveness, the mean activity of both tapping 3–5 and tapping 21–30 was subtracted from *Act1 tap* to obtain *Diff 3–5 vs 1 tap* and *Diff 21–30 vs 1 tap*, respectively. For the thigmotaxis parameters, arenas were split into center and periphery (38.5 mm$^2$ each), activity (% delta pixel) and total distance moved (TDM) (mm) were exported by minute and arena (periphery and center) using EthoVisionXT. To exclude larvae with suboptimal positional tracking activity (pixel-based) and TDM (tracking-based), larvae with a Pearson correlation of $r<0.9$ were removed. Larvae with little movement (activity = 0 for<3 min per condition) were also excluded. Trials with <3 larvae in any genotype were removed. Median thigmotaxis values per larvae were computed as TDM (mm) in the periphery divided by the sum of TDM (mm) in the center and periphery, multiplied by 100, for the time intervals of B, D, and L only considering minutes of movement only (activity >0).

## Evaluation of social interactions in juvenile fish

### Implementation of a custom behavioral system

The behavioral setup developed in the lab was based on *Hinz and de Polavieja, 2017*, developed by the Polavieja lab. It consisted of a custom-made, circular transparent tank with a diameter of 42 cm, filled with 4 L of system water (*Figure 5—figure supplement 1*). This experimental arena was placed in the center of an opaque acrylic sheet held within a larger external tank measuring 80x60 x 20 cm, which contained 90 L of system water. To maintain a consistent water temperature of 28 °C, two water heaters were placed diagonally opposite each other in the external tank. The entire setup was housed in a 1.2 m² box with white walls. Uniform illumination was crucial, so LEDs were installed around the inside of the box. A hand-sewn curtain was used as a light diffuser, covering the diameter of the experimental arena to prevent reflections. The overall light intensity reached 2000 Lux. Videos

were recorded from above using a 12.36 MP UI-3000SE monochrome camera with a 16 mm Tamron lens. The camera was mounted on an adjustable arm, positioned 100 cm above the center of the tank. Video recordings were conducted using Stream-Pix 7 software, running on a high-performance computer equipped with an Nvidia GeForce GTX 1070–8 G graphics card for data acquisition and post-processing.

## Experimental design and procedures

We used juvenile zebrafish at approximately 30 dpf, conducting experiments over two consecutive days for each microexon mutant line to ensure a representative sample size. Due to the slower growth rate of the homozygous *srrm3* mutant fish, they were studied at 3 mpf, when their body length matched that of 30 dpf wild-type zebrafish. Sex was not considered, as it is undetermined at this stage. All experiments were conducted between 8:00 AM and 4:00 PM. We used homozygous × heterozygous crosses to simplify the design and avoid an excessive number of genotype combinations. Each experiment was performed blind to fish pairs, and animals were fin-clipped and genotyped post-experiment. On the day of the experiment, fish tanks were placed in the experimental box for 1 hr to acclimate the animals. The experimental protocol had the following steps (*Figure 5A*): (i) selecting a random pair of fish and introducing them to the arena, (ii) allowing a 2 min habituation, (iii) recording free movement for 10 min, (iv) removing fish 1, (v) removing fish 2 one minute later, (vi) anesthetizing, (vii) genotyping. After removal, fish were placed in a 1 L fish water tank with 30 mL of tricaine (4 mg/mL, Sigma-E10521) for fin clipping, a critical step for matching each fish to its experimental ID, preventing misidentification. Videos were recorded at 30 frames per second (fps) using Stream-Pix 7, with manual image adjustments.

## Video tracking with idtracker.ai

After video acquisition, the recordings were compressed using FFmpeg (version 8), an open-source multimedia framework. This reduced the file size while maintaining acceptable visual quality. The compressed files were successfully validated by tracking them alongside the uncompressed versions to ensure they met the required quality for further analysis. We used idtracker.ai (*Romero-Ferrero et al., 2019*) to track animals and extract the trajectories of the fish in each video. These trajectories form the basis for analyzing animal behavior and interactions parameters. Before running the tracking, specific parameters were adjusted individually for each video using the graphical user interface (GUI) of idtracker.ai to ensure proper identification of the fish. The key parameters were: (i) Region of Interest (ROI): adjusted to match the perimeter of the arena (*Figure 5—figure supplement 1*), (ii) Time intervals: first interval set to 10 min (18,000 frames)+second interval of variable time for fish identification, (iii) Pixel intensity: used to differentiate fish from the background, ranging from 0 to 255, (iv) Area threshold: defined between 0 and 6000 to identify individual fish blobs by filtering based on size (and excluding other objects of different sizes), allowing for better distinction from pixel intensity variations. (v) Number of blobs: set to 2 (one for each fish in the pair). After setting these parameters, all videos were tracked, and tracking performance was evaluated based on the estimated accuracy, which measures the proportion of correctly identified fish. An accuracy threshold of 0.98 was applied, allowing for up to 2% misidentification. Videos falling below this accuracy threshold were excluded from further analysis.

## Data processing and statistical analysis

For data processing, we customized the Python-based *trajectorytools* package (https://github.com/fjhheras/trajectorytools, copy archived at *Rance et al., 2024*) to better suit our needs (https://github.com/vastgroup/mic-social-behavior-analysis, copy archived at *Ferrero, 2022*). First, we included 'Genotype' (Heterozygous or Homozygous) and 'Pair' (Het-Het, Het-Del, Del-Del) as key variables for statistical comparisons. Next, we developed an automated system to associate each fish with its experimental ID and correlate it with the tracking ID. We leveraged the video's second interval, which captures the removal of the two fish, to define the experimental ID for genotyping. By checking for the presence or absence of trajectories in this interval, we matched each fish to its corresponding trajectory and ID number. We used *trajectorytools* to import the trajectories generated by idtracker.ai and to compute the following variables from the fish trajectories: (i) Speed: rate at which an individual fish or group of fish moves through their environment measured in cm/s. (ii) Absolute normal acceleration

(*Abs. N. Accel): absolute value of the* acceleration vector component that is directly perpendicular to the fish's current direction of movement. This component of the acceleration is associated with changes in direction (turning) and is measured in cm/s². (iii) Absolute tangential acceleration (*Abs. Tg. Accel):* absolute value of the acceleration vector that is parallel to the fish's current direction of movement. It represents how fast the zebrafish is speeding up or slowing down without changing direction (associated escape response and group cohesion) measured in cm/s². (iv) Distance traveled*:* distance traveled (cm). (v) Time in the periphery: Total number of frames that a fish was found in the periphery (border) of the arena. We defined the periphery as the ring defined from 0.8*R to the border, where R is the radius of the arena. (vi) Normalized distance to origin: Normalized distance to the center of the arena being 0=center and 1=border (walls). (vii) Ratio in front: The proportion of time that a fish has another fish in front. From a focal fish perspective, lower values are associated with higher leadership. (viii) Polarization: measure of how aligned the zebrafish are in terms of their movement direction. When polarization is low, individuals are moving in random directions. When polarization is high, most individuals in the group are moving in the same or similar directions. Unitless, range 0–1. (ix) Distance to neighbor: distance between the two fish measured in cm.

Measures i-vii corresponded to individual fish measures within a pair, while measures viii and ix are pair-specific. For statistical analyses, for the individual measures, we split them by genotype (heterozygous [Het] or homozygous [Del]) and according to the pair in which they were tested (homotypic Het-Het or Del-Del [same, S], heterotypic Het-Del [different, D]). For measures i-vi, a single value for each S pair was collected (the average of the two fish), to avoid double counting of nearly identical measures. For group features, we split them by pair type and compared Het-Het pairs against either Het-Del or Del-Del. For all comparisons, two-sided Wilcoxon Rank-Sum tests were used to assess statistically significant differences.

## Considerations regarding statistical analyses

Given the complex and diverse structure of the data in *Figures 3–5*, as well as the associated supplemental figures, we took the following approaches. When a varying number of biological replicates of the same founder were present (e.g. *Figures 3C and 4B*), we performed a permutation-based test to obtain the median of the p-value distribution of bootstrap resampling Wilcoxon tests. When the two founders were tested together (e.g. *Figure 3D*), we used ANOVA tests to assess the significance of genotype. Finally, for direct comparisons between the distributions of specific features between WT and Het/Del, we used two-sided Wilcoxon Rank-Sum tests (*Figures 4 and 5* and boxplots in the associated figure supplementary figures). Importantly, given the large amount of tests performed in those figures, and given that defining the level to which to apply the multiple testing correction (microexon, feature, figure, etc.) would be to a large extent arbitrary, we opted for reporting the nominal p-value and using this value not as a hard cut-off, but as a way to help identify and highlight potentially meaningful biological effects, along with the effect size of the difference and the consistency across replicates and founders (when available).

## Generation of RNA sequencing and transcriptomic analysis

To assess transcriptome-wide changes in gene expression, we generated RNA-seq for WT and Del 5 dpf larvae for all 18 microexon main founder lines, as well as 5 secondary ones (*Supplementary file 6*). In addition, we generated two biological replicates of *srrm3* WT and Del 5 dpf larvae. To produce these data, incrosses of heterozygous parents were set for each line, and larvae were genotyped by fin clipping (see above) at 5 dpf. Between 10–12 larvae of either WT or Del were pooled for each line and RNA was extracted using the RNeasy Mini Kit (QIAGEN, 74134), following the manufacturer's instructions. Final RNA concentrations were quantified using a Qubit Fluorometer with the RNA HS Assay Kit (Invitrogen, Q32855). PolyA +stranded libraries for RNA sequencing were generated at the CRG Genomics Unit and sequenced in an Illumina HiSeq2500 machine to produce an average of 42.5 million 50-nt single-end reads (*Supplementary file 6*). For samples of the regulator mutants, 72.8 million 125-nt paired-end reads were generated to allow for alternative splicing analyses. All RNA-seq samples were submitted to Gene Expression Omnibus (GEO), identifier: GSE278690.

RNA-seq data was mapped with STAR (*Dobin et al., 2013*) to the *Danio rerio* genome version danRer10 (Ensembl v91). Mappings were used for correcting bias due to the heterogeneous patterns of transcript degradation with the Bioconductor package named *DegNorm* (*Xiong et al., 2019*)

using the gene annotations from Ensembl v91. Gene read counts were then normalized with the variance stabilizing transformation (VST) from DESeq2 (*Love et al., 2014*). Additionally, batch variation was removed using the Empirical Bayes-moderate adjustment for unwanted covariates from the R package WGCNA fitting WT samples (*Figure 6—figure supplement 1*). Differences between Del and WT samples were obtained from the subtraction of these values from each genotype after all corrections and normalizations. Gene Set Enrichment Analyses (GSEA) were performed with the R package *fgsea* using *idep* (*Ge et al., 2018*) and *shinyGO* (*Ge et al., 2020*) GO annotations. To select globally enriched GO categories across microexon deletion lines, we first sum the absolute differences between Del and WT across all main founders and performed a positive GSEA test. Jaccard distance was then calculated between enriched GOs (adjusted *P*-value <0.01) and used for a complete hierarchical clustering, and tree cutting at a 0.9 height (<10% shared genes between clustered GOs). Clusters with at least one term with a p-value <0.001 were filtered and named after the most significant term in it. Additionally, for each individual microexon deletion comparison, a GSEA was estimated for GOs enriched globally and merged cluster categories (*Figure 6C*, right plot). Finally, we selected various gene sets of interest to investigate their differential expression patterns among microexon deletions using GSEA (*Figure 6D*). These gene sets (*Supplementary file 7*) were obtained as follows: (i) ASD genes: zebrafish orthologs of SFARI category 1 and 2 genes [April 2020]; (ii) Schizophrenia genes: from *Thyme et al., 2019*; (iii) Immediate early genes: zebrafish orthologs of genes reported in *Piwecka et al., 2017*; (iv) Developmental pathways: from *Marlétaz et al., 2018*.

## Acknowledgements

This project has received funding from the European Research Council (ERC) under the European Union's Horizon 2020 research and innovation programme (grant agreements No 637591 and No 101002275, both to MI), the Spanish Ministry of Science, Innovation and Universities through the State Investigation Agency (PID2020-115040GB-I00/AEI / 10.13039/501100011033 to MI) and the AGAUR (2021 SGR-Cat 01220). CRG acknowledges support of the Spanish Ministry of Science and Innovation through the Centro de Excelencia Severo Ochoa (CEX2020-001049-S, MCIN/ AEI/10.13039/501100011033), and the Generalitat de Catalunya through the CERCA program. GP and FRF were funded by Champalimaud Foundation and the FCT (PTDC/BIA-COM/5770/2020). RNA-seq samples were generated at the CRG Genomics Unit and NanoBRET assays at the CRG Protein Technologies Unit, both part of the CRG Core Technologies Programme.

## Additional information

### Funding

| Funder | Grant reference number | Author |
|---|---|---|
| European Commission | 10.3030/637591 | Manuel Irimia |
| European Commission | 10.3030/101002275 | Manuel Irimia |
| Agencia Estatal de Investigación | PID2020-115040GB-I00/ AEI/10.13039/501100011033 | Manuel Irimia |
| Agència de Gestió d'Ajuts Universitaris i de Recerca | 2021 SGR-Cat 01220 | Manuel Irimia |
| Spanish Ministry of Science and Innovation | CEX2020-001049-S | Manuel Irimia |
| Fundação para a Ciência e Tecnologia | PTDC/BIA-COM/5770/2020 | Gonzalo de Polavieja |
| Spanish Ministry of Science and Innovation | MCIN/ AEI/10.13039/501100011033 | Manuel Irimia |
| Consejeria de Educación, Junta de Castilla y León | CSI018P23 | Myriam Cuadrado Xosé R Bustelo |

| Funder | Grant reference number | Author |
|---|---|---|
| Agencia Estatal de Investigación | PID2021-122666OB-I00 | Myriam Cuadrado<br>Xosé R Bustelo |
| Agencia Estatal de Investigación | PID2024-156980OB-I00 | Xosé R Bustelo |
| Asociación Española Contra el Cáncer | EPAEC222641CICS | Xosé R Bustelo |
| Consejería de Educación, Junta de Castilla y León | CLU-2023-2-01 | Xosé R Bustelo |

The funders had no role in study design, data collection and interpretation, or the decision to submit the work for publication.

### Author contributions

Laura Lopez-Blanch, Data curation, Formal analysis, Investigation, Visualization, Methodology; Cristina Rodríguez-Marin, Elizabeth M Kita, Myriam Cuadrado, Investigation, Methodology; Federica Mantica, Tahnee Mackensen, Software, Investigation, Visualization; Luis P Iñiguez, Software, Investigation, Visualization, Methodology; Jon Permanyer, Data curation, Validation, Investigation; Mireia Codina-Tobias, Investigation; Francisco Romero-Ferrero, Data curation, Software, Visualization; Jordi Fernandez-Albert, Methodology; Xosé R Bustelo, Resources, Supervision, Funding acquisition; Gonzalo de Polavieja, Resources, Software, Supervision; Manuel Irimia, Conceptualization, Resources, Formal analysis, Supervision, Funding acquisition, Investigation, Writing – original draft, Project administration, Writing – review and editing

### Author ORCIDs

Cristina Rodríguez-Marin ⬮ https://orcid.org/0000-0001-6772-4853
Luis P Iñiguez ⬮ https://orcid.org/0000-0002-1470-9024
Tahnee Mackensen ⬮ https://orcid.org/0000-0002-6537-7592
Mireia Codina-Tobias ⬮ https://orcid.org/0009-0003-6005-270X
Myriam Cuadrado ⬮ https://orcid.org/0000-0001-9410-1205
Xosé R Bustelo ⬮ https://orcid.org/0000-0001-9398-6072
Gonzalo de Polavieja ⬮ https://orcid.org/0000-0001-5359-3426
Manuel Irimia ⬮ https://orcid.org/0000-0002-2179-2567

### Ethics

Fish procedures were approved by the Institutional Animal Care and Use Ethic Committee (PRBB-IACUEC).

Reviewer #1 (Public review): https://doi.org/10.7554/eLife.104275.3.sa1
Reviewer #3 (Public review): https://doi.org/10.7554/eLife.104275.3.sa2
Author response https://doi.org/10.7554/eLife.104275.3.sa3

---

## Additional files

### Supplementary files

Supplementary file 1. Neural exons and microexons in zebrafish.

Supplementary file 2. Selected microexons for phenotypic characterization and associated regulatory data.

Supplementary file 3. Sequence of guide RNAs used to generate microexon deletion lines.

Supplementary file 4. Primers for genotyping and for mRNA validation of microexon inclusion.

Supplementary file 5. Config file used for the *Get_Tissue_Specific_AS.pl* script.

Supplementary file 6. RNA-seq samples generated in this study.

Supplementary file 7. Gene sets of special interest used in this study.

MDAR checklist

Source data 1. Spreadsheet containing all the numerical values used to generate the plots in each

Source data 2. Raw data for the figures.

figure panel not covered in *Source code 1* and *Source data 2*.

Source code 1. Code used to generate the plots from the raw data of *Source data 2*.

### Data availability

RNA sequence data were deposited in GEO (accession number GSE278690). All data generated or analyzed during this study are included in the manuscript and supporting files; source data files have been provided for all figures. Relevant software is provided in *Source code 1* (along with the necessary raw data in *Source data 2*) or in https://github.com/vastgroup/mic-social-behavior-analysis (*Romero-Ferrero, 2022*). *Source data 1* contains the remaining numerical data used to generate the figures not covered in Source code 1 and Source code 2. Raw, uncropped western blot gel images corresponding to Figure 3I are provided as *Figure 3—source data 1* and *Figure 3—source data 2*. Raw, uncropped gel images underlying the RT-PCR assays presented in Figure 1—figure supplement 1, Figure 2—figure supplement 1 and Figure 2C are unavailable as source data. The original files were stored on the local computer of a former laboratory member who departed in 2023, and no backup existed when the device was decommissioned during an institutional relocation in January 2024.

The following datasets were generated:

| Author(s) | Year | Dataset title | Dataset URL | Database and Identifier |
|---|---|---|---|---|
| Lopez-Blanch L, Rodriguez-Marin C, Irimia M | 2025 | Phenotypic impact of individual conserved neuronal microexons and their master regulators in zebrafish | https://doi.org/10.17632/3b3zx4cfg9.1 | Mendeley Data, 10.17632/3b3zx4cfg9.1 |
| Martinez-Ordoñez A, Duran A, Ruiz-Martinez M, Cid-Diaz T, Zhang X, Han Q, Kinoshita H, Muta Y, Linares JF, Kasashima H, Omar M, Yashiro M, Pannellini T, Pigazzi A, Inghirami G, Marchionni L, Sigal D, Diaz-Meco MT, Moscat J | 2022 | Hyaluronan driven by epithelial aPKC deficiency remodels the microenvironment and creates a therapeutic vulnerability in mesenchymal colorectal cancer IV | https://www.ncbi.nlm.nih.gov/geo/query/acc.cgi?acc=GSE207780 | NCBI Gene Expression Omnibus, GSE207780 |

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

# Appendix 1

**Appendix 1—key resources table**

| Reagent type (species) or resource | Designation | Source or reference | Identifiers | Additional information |
|---|---|---|---|---|
| Strain, strain background (*Danio rerio*) | Tg(HuC:GFP) | *Park et al., 2000* | Tg(HuC:GFP) | |
| Strain, strain background (*Danio rerio*) | Tg(HuC:GFP; Gli3_EX0035009_9) | This paper | gli3_9 | |
| Strain, strain background (*Danio rerio*) | Tg(HuC:GFP; Gli3_EX0035009_7) | This paper | gli3_7 | |
| Strain, strain background (*Danio rerio*) | Tg(HuC:GFP; Apbb1_EX0013846_1) | This paper | apbb1_1 | |
| Strain, strain background (*Danio rerio*) | Tg(HuC:GFP; Mef2ca_EX0045884_2) | This paper | mef2ca_2 | |
| Strain, strain background (*Danio rerio*) | Tg(HuC:GFP; Evi5b_EX0030978_3) | This paper | evi5b_3 | |
| Strain, strain background (*Danio rerio*) | Tg(HuC:GFP; Evi5b_EX0030978_6) | This paper | evi5b_6 | |
| Strain, strain background (*Danio rerio*) | Tg(HuC:GFP; Ap1g1_EX0013683_1) | This paper | ap1g1_1 | |
| Strain, strain background (*Danio rerio*) | Tg(HuC:GFP; Vdac3_EX0084235_5) | This paper | vdac3_5 | |
| Strain, strain background (*Danio rerio*) | Tg(HuC:GFP; kif1b_EX0041536_1) | This paper | kif1b_1 | |
| Strain, strain background (*Danio rerio*) | Tg(HuC:GFP; kif1b_EX0041536_2) | This paper | kif1b_2 | |
| Strain, strain background (*Danio rerio*) | Tg(HuC:GFP; Itsn1_EX0040386_1) | This paper | itsn1_1 | |
| Strain, strain background (*Danio rerio*) | Tg(HuC:GFP; Itsn1_EX0040386_3) | This paper | itsn1_3 | |
| Strain, strain background (*Danio rerio*) | Tg(HuC:GFP; Cdon_EX0020538_1) | This paper | cdon_1 | |
| Strain, strain background (*Danio rerio*) | Tg(HuC:GFP; Asap1b_EX0015073_3) | This paper | asap1b_3 | |
| Strain, strain background (*Danio rerio*) | Tg(HuC:GFP; Asap1b_EX0015073_7) | This paper | asap1b_7 | |
| Strain, strain background (*Danio rerio*) | Tg(HuC:GFP; Vav2_EX008416_1) | This paper | vav2_1 | |
| Strain, strain background (*Danio rerio*) | Tg(HuC:GFP; Vav2_EX008416_2) | This paper | vav2_2 | |

*Appendix 1 Continued on next page*

*Appendix 1 Continued*

| Reagent type (species) or resource | Designation | Source or reference | Identifiers | Additional information |
|---|---|---|---|---|
| Strain, strain background (*Danio rerio*) | Tg(HuC:GFP; Shank3b_EX0066093_1) | This paper | shank3b_1 | |
| Strain, strain background (*Danio rerio*) | Tg(HuC:GFP; Shank3b_EX0066093_2) | This paper | shank3b_2 | |
| Strain, strain background (*Danio rerio*) | Tg(HuC:GFP; Src_EX0075047_1) | This paper | src_1 | |
| Strain, strain background (*Danio rerio*) | Tg(HuC:GFP; Src_EX0075047_2) | This paper | src_2 | |
| Strain, strain background (*Danio rerio*) | Tg(HuC:GFP; Gli2b_EX0035002_1) | This paper | gli2b_1 | |
| Strain, strain background (*Danio rerio*) | Tg(HuC:GFP; Gli2b_EX0035002_2) | This paper | gli2b_2 | |
| Strain, strain background (*Danio rerio*) | Tg(HuC:GFP; Madd_EX0044304_1) | This paper | madd_1 | |
| Strain, strain background (*Danio rerio*) | Tg(HuC:GFP; Madd_EX0044304_2) | This paper | madd_2 | |
| Strain, strain background (*Danio rerio*) | Tg(HuC:GFP; Vti1a_EX0084665_1) | This paper | vti1a_1 | |
| Strain, strain background (*Danio rerio*) | Tg(HuC:GFP; Vti1a_EX0084665_2) | This paper | vti1a_2 | |
| Strain, strain background (*Danio rerio*) | Tg(HuC:GFP; Shank3a_EX0066086_1) | This paper | shank3a_1 | |
| Strain, strain background (*Danio rerio*) | Tg(HuC:GFP; Shank3a_EX0066086_2) | This paper | shank3a_2 | |
| Strain, strain background (*Danio rerio*) | Tg(HuC:GFP; Reln_EX0007444_1) | This paper | reln_1 | |
| Strain, strain background (*Danio rerio*) | Tg(HuC:GFP; Reln_EX0007444_2) | This paper | reln_2 | |
| Antibody | mouse anti-acetylated tubulin | BioLegend | 802001 | 1:400 |
| Antibody | rabbit anti-GFP | Millipore | AB3080P | 1:500 |
| Antibody | Alexa Fluor 594 goat anti-mouse | Thermo Fisher | A-11005 | 1:500 |
| Antibody | Alexa Fluor 488 goat anti-rabbit | Thermo Fisher | A-11034 | 1:500 |
| Antibody | Hoechst | Sigma | B2261 | 1:1000 |
| Antibody | anti-3A10 | Developmental Studies Hybridoma Bank | AB_531874 | |
| Antibody | anti-GFP | Covance, Princeton, NJ, USA | MMS-118P | 1:2000 |

*Appendix 1 Continued on next page*

*Appendix 1 Continued*

| Reagent type (species) or resource | Designation | Source or reference | Identifiers | Additional information |
|---|---|---|---|---|
| Antibody | anti-β-actin | Santa Cruz | sc-47778 | 1:1000 |
| Strain, strain background (*Danio rerio*) | srrm3 | *Ciampi et al., 2022* | srrm3_17 | |
| Strain, strain background (*Danio rerio*) | srrm4 | *Ciampi et al., 2022* | srrm4_12D | |
| Gene (*Danio rerio*) | gli3_mic | VastDB | EX0035009 | |
| Gene (*Danio rerio*) | mef2ca_mic | VastDB | EX0045884 | |
| Gene (*Danio rerio*) | evi5b_mic | VastDB | EX0030978 | |
| Gene (*Danio rerio*) | ap1g1_mic | VastDB | EX0030978 | |
| Gene (*Danio rerio*) | vdac3_mic | VastDB | EX0084235 | |
| Gene (*Danio rerio*) | kif1b_mic | VastDB | EX0041536 | |
| Gene (*Danio rerio*) | itsn1_mic | VastDB | EX0040386 | |
| Gene (*Danio rerio*) | apbb1_mic | VastDB | EX0013846 | |
| Gene (*Danio rerio*) | cdon_mic | VastDB | EX0020538 | |
| Gene (*Danio rerio*) | asap1b_mic | VastDB | EX0015073 | |
| Gene (*Danio rerio*) | vav2_mic | VastDB | EX008416 | |
| Gene (*Danio rerio*) | shank3b_mic | VastDB | EX0066093 | |
| Gene (*Danio rerio*) | src_mic | VastDB | EX0075047 | |
| Gene (*Danio rerio*) | gli2b_mic | VastDB | EX0035002 | |
| Gene (*Danio rerio*) | madd_mic | VastDB | EX0044304 | |
| Gene (*Danio rerio*) | vti1a_mic | VastDB | EX0084665 | |
| Gene (*Danio rerio*) | shank3a_mic | VastDB | EX0066086 | |
| Gene (*Danio rerio*) | reln_mic | VastDB | EX0007444 | |
| Gene (*Danio rerio*) | srrm3 | Ensembl | ENSDARG00000096920 | |
| Gene (*Danio rerio*) | srrn4 | Ensembl | ENSDARG00000086327 | |
| Cell line (*Chlorocebus aethiops*) | COS-7 | Created by Yakov Gluzman in the early 1980s. | NA | Cell line maintained in X. Bustelo lab; male. |
| Transfected construct (*Homo sapiens*) | PathDetect SRF cis-Reporting System | Agilent | 219081 | |
| Transfected construct (NA) | pRL-SV40 | Promega | E2231 | Renilla luciferase |
| Transfected construct (*Homo sapiens*) | VAV2_onco | *Lorenzo-Martín et al., 2020* | Positive control VAV2 activity | pNM115 |
| Transfected construct (*Danio rerio*) | VAV2_MIC+ | This paper | | Zebrafish vav2 cDNA with microexon |
| Transfected construct (*Danio rerio*) | VAV2_MIC- | This paper | | Zebrafish vav2 cDNA without microexon |

