## [Editor Report · eLife Assessment]

This **important** work examines how microexons contribute to brain activity, structure, and behavior. The authors find that loss of microexon sequences generally has subtle impacts on these metrics in larval zebrafish, with few exceptions. The evidence is **convincing**, using modern high-throughput phenotyping methodology in zebrafish. Overall, this work will be of interest to neuroscientists and generate further studies of interest to the field.

---

## [Referee Report · Reviewer #1 (Public review)]

Summary:

In this manuscript by Lopez-Blanch and colleagues, 21 microexons are selected for a deep analysis of their impacts on behavior, development, and gene expression. The authors begin with a systematic analysis of microexon inclusion and conservation in zebrafish and use these data to select 21 microexons for further study. The behavioral, transcriptomic, and morphological data presented are largely convincing and discussion of the potential explanations for the subtle impacts of individual microexon deletions versus loss-of-function in srrm3 and/or srrm4 is quite comprehensive and thoughtful.

Strengths:

The study uses a wide variety of techniques to assess the impacts of microexon deletion, ranging from assays of protein function through regulation of behavior and development.

The authors provide comprehensive analyses of the molecular impact of their microexon deletions, including examining how host-gene and paralog expression is affected.

---

## [Referee Report · Reviewer #3 (Public review)]

Summary:

Microexons are highly conserved alternative splice variants, the individual functions of which have thus far remained mostly elusive. Inclusion of microexons in mature mRNAs increases during development, specifically in neural tissues, and is regulated by SRRM proteins. Investigation of individual microexon function is a vital avenue of research, since microexon inclusion is disrupted in diseases like autism. This study provides one of the first rigorous screens (using zebrafish larvae) of the functions of individual microexons in neurodevelopment and behavioural control. The authors precisely excise 21 microexons from the genome of zebrafish using CRISPR-Cas9 and assay the downstream impacts on neurite outgrowth, larvae motility and sociality. A small number of mild phenotypes were observed, which contrasts with the more dramatic phenotypes observed when microexon master regulators SRRM3/4 are disrupted. Importantly, this study attempts to address the reasons why mild/few phenotypes are observed and identifies transcriptomic changes in microexon mutants that suggest potential compensatory gene regulatory mechanisms.

Strengths:

(1) The manuscript is well written with excellent presentation of the data in the figures.

(2) The experimental design is rigorous and explained in sufficient detail.

(3) The identification of a potential microexon compensatory mechanism by transcriptional alterations represents a valued attempt to begin to explain complex genetic interactions.

Overall this is a study with robust experimental design that addresses a gap in knowledge of the role of microexons in neurodevelopment.

---

## [Author Response]

The following is the authors’ response to the original reviews.

**Reviewer #1 (Public review):**
Summary:In this manuscript by Lopez-Blanch and colleagues, 21 microexons are selected for a deep analysis of their impacts on behavior, development, and gene expression. The authors begin with a systematic analysis of microexon inclusion and conservation in zebrafish and use these data to select 21 microexons for further study. The behavioral, transcriptomic, and morphological data presented are for the most part convincing. Furthermore, the discussion of the potential explanations for the subtle impacts of individual microexon deletions versus lossof-function in srrm3 and/or srrm4 is quite comprehensive and thoughtful. One major weakness: data presentation, methods, and jargon at times affect readability / might lead to overstated conclusions. However, overall this manuscript is well-written, easy to follow, and the results are of broad interest.

We thank the Reviewer for their positive comments on our manuscript. In the revised version, we will try to improve readability, reduce jargon and avoid overstatements.

Strengths:(1) The study uses a wide variety of techniques to assess the impacts of microexon deletion, ranging from assays of protein function to regulation of behavior and development.(2) The authors provide comprehensive analyses of the molecular impact of their microexon deletions, including examining how host-gene and paralog expression is affected.Weaknesses:Major Points:(1) According to the methods, it seems that srrm3 social behavior is tested by pairing a 3mpf srrm3 mutant with a 30dpf srrm3 het. Is this correct? The methods seem to indicate that this decision was made to account for a slower growth rate of homozygous srrm3 mutant fish. However, the difference in age is potentially a major confound that could impact the way that srrm3 mutants interact with hets and the way that srrm3 mutants interact with one another (lower spread for the ratio of neighbour in front value, higher distance to neighbour value). This reviewer suggests testing het-het behavior at 3 months to provide age-matched comparisons for del-del, testing age-matched rather than size-matched het-del behavior, and also suggests mentioning this in the main text / within the figure itself so that readers are aware of the potential confound.

Thank you for bringing up this point. For the tests shown in Figure 5, we indeed decided to match the pairs involving srrm3 mutant fish by fish size since we reasoned this would be more comparable to the other lines, both biologically and methodologically (in terms of video tracking, etc.). However, we are confident the results would be very similar if matched by age, since the differences in social interactions between the srrm3 homozygous mutants and their control siblings are very dramatic at any age. As an example, this can be appreciated, in line with the Reviewer's suggestion, in Videos S2 and S3, which show groups of five 5 mpf fish that are either srrm3 mutant or wild type. It can be observed that the behavior of 5 mpf WT fish (Video S3) is very similar to those of 1 mpf WT fish pairs, with very small interindividual distances, while the difference with repect to the srrm3 mutant group (Video S2) is dramatic. We nonetheless agree that this decision on the experimental design should be clearly stated in the main text and figure legend and we have done so in the revised version.

(2) Referring to srrm3+/+; srrm4-/- controls for double mutant behavior as "WT for simplicity" is somewhat misleading. Why do the authors not refer to these as srrm4 single mutants?

This comment applies to Figure 4 as well as the associated figure supplements. We reasoned that this made the understanding of plots easier, but the Reviewer is correct that it can be misleading. As a middle ground, we have now changed Figure 4 to follow the nomenclature of Figure 3D (WD, HD, DD), which is further explained in the legend, but kept the original format in the figure supplements for consistency with the (many) other plots in those figures.

(3) It's not completely clear how "neurally regulated" microexons are defined / how they are different from "neural microexons"? Are these terms interchangeable?

Yes, they are interchangeable. We have now double checked the wording to avoid confusion and for consistency.

(4) Overexpression experiments driving srrm3 / srrm4 in HEK293 cells are not described in the methods.

We apologized for this omission. We now briefly describe the data and asscoiated methods in more detail in the revised version; however, please note that the data was obtained from a previous publication (Torres-Mendez et al, 2019), where the detailed methodology is reported.

(5) Suggest including more information on how neurite length was calculated. In representative images, it appears difficult to determine which neurites arise from which soma, as they cross extensively. How was this addressed in the quantification?

We have added further details to the revised version. With regards to the specific question, we would like to mention that this has not been a very common issue for the time points used in the manuscript (10 hap and 24 hap). At those stages, it was nearly always evident how to track each individual neurite. Dubious cases were simply ignored and not measured, as we aimed for 100 neurites per well. Of course, such complex cases become much more common at later time points (48 and 72 hap), which were not used in this study.

**Reviewer #2 (Public review):**
Summary:This manuscript explores in zebrafish the impact of genetic manipulation of individual microexons and two regulators of microexon inclusion (Srrm3 and Srrm4). The authors compare molecular, anatomical, and behavioral phenotypes in larvae and juvenile fish. The authors test the hypothesis that phenotypes resulting from Srrm3 and 4 mutations might in part be attributable to individual microexon deletions in target genes.The authors uncover substantial alterations in in vitro neurite growth, locomotion, and social behavior in Srrm mutants but not any of the individual microexon deletion mutants. The individual mutations are accompanied by broader transcript level changes which may resemble compensatory changes. Ultimately, the authors conclude that the severe Srrm3/4 phenotypes result from additive and/or synergistic effects due to the de-regulation of multiple microexons.Strengths:The work is carefully planned, well-described, and beautifully displayed in clear, intuitive figures. The overall scope is extensive with a large number of individual mutant strains examined. The analysis bridges from molecular to anatomical and behavioral read-outs. Analysis appears rigorous and most conclusions are well-supported by the data.Overall, addressing the function of microexons in an in vivo system is an important and timely question.Weaknesses:The main weakness of the work is the interpretation of the social behavior phenotypes in the Srrm mutants. It is difficult to conclude that the mutations indeed impact social behavior rather than sensory processing and/or vision which precipitates apparent social alterations as a secondary consequence. Interpreting the phenotypes as "autism-like" is not supported by the data presented.

The Reviewer is absolutely right. It was not our intention to imply that these social defects should be interpreted simply as autistic-like. It is indeed very likely that the main reason for the social alterations displayed by the srrm3 mutants is their impaired vision. We have now added this discussion point explicitly in the revised version.

**Reviewer #3 (Public review):**
Summary:Microexons are highly conserved alternative splice variants, the individual functions of which have thus far remained mostly elusive. The inclusion of microexons in mature mRNAs increases during development, specifically in neural tissues, and is regulated by SRRM proteins. Investigation of individual microexon function is a vital avenue of research since microexon inclusion is disrupted in diseases like autism. This study provides one of the first rigorous screens (using zebrafish larvae) of the functions of individual microexons in neurodevelopment and behavioural control. The authors precisely excise 21 microexons from the genome of zebrafish using CRISPR-Cas9 and assay the downstream impacts on neurite outgrowth, larvae motility, and sociality. A small number of mild phenotypes were observed, which contrasts with the more dramatic phenotypes observed when microexon master regulators SRRM3/4 are disrupted. Importantly, this study attempts to address the reasons why mild/few phenotypes are observed and identify transcriptomic changes in microexon mutants that suggest potential compensatory gene regulatory mechanisms.Strengths:(1) The manuscript is well written with excellent presentation of the data in the figures.(2) The experimental design is rigorous and explained in sufficient detail.(3) The identification of a potential microexon compensatory mechanism by transcriptional alterations represents a valued attempt to begin to explain complex genetic interactions.(4) Overall this is a study with a robust experimental design that addresses a gap in knowledge of the role of microexons in neurodevelopment.

Thank you very much for your positive comments to our manuscript.

**Reviewer #1 (Recommendations for the authors):**
Minor Suggestions(1) Axes are often scaled differently even between panels in the same figure. For example in Figure 5 - supplement 10, the srrm3_17 y axis scales from 0-20, while the neighboring panels scale from ~1-2.5. This somewhat underrepresents the finding that srrm3 mutants have much larger inter-individual distances. Similarly, in the panel above (src_1), the y-axis is scaled to include a single point around 17cm. As a result, it appears at first glance that the src_1 trials resulted in much lower inter-individual distance. Suggest scaling all of these the same to improve readability.

While the Reviewer is certainly correct, after careful consideration we decided to have autoscaled axis to prioritize within-plot visualization (i.e. among genotypes within an experiment) than across plots (i.e. among experiments and lines).

(2) Attention to italicizing gene names.

Thanks.

(3) In many points in the methods, we are instructed to "see below." Suggest directing the reader to a particular section heading.

We found only one such instance, and we directed the reader to the specific section, as suggested.

(4) In Methods, remove "in the corpus callosum." This is not an accurate descriptor for the site at which Mauthner axons cross.

This is absolutely correct, apologies for this mistake.

Clarify:(1) In the results section, "tissue-specific regulation was validated..." - suggest mentioning that this was performed in adult tissues / describe dissection in the methods.

Added.

(2) In the results section, the meaning of "no event ortholog" is not clear. Does this mean that a microexon does not have a human homolog? If so, suggest stating more clearly.

Correct. We have added addition information.

(3) In the results, the authors state that 78% of microexons are affected by srrm3/4 loss-offunction. Suggest stating the method used here (e.g. RNA-seq in mutants as compared to siblings)

Added.

(4) It is not clear what "siblings for the main founders means" for example in 3D. Is this effectively the analysis of microexon knockouts across multiple independent lines? Are the lines pooled for stats, for example in 3C?

The main founder correspond to that listed as _1 and as default for experiments when only one found is used. We now explicitely state this.

For 3C, the lines are not pooled for stats; the stats correspond only to the main founder for each line. However, for each main founder line, multiple experiments are usually analyzed together and the stats are done taking their data structure into account (i.e. not simply pooling the values).

(5) The purpose and a general description of NanoBRET assays should be included in the results.

We added the main purpose of the NanoBRET assays (testing protein-protein interactions).

(6) Specify that baseline behavior is analyzed in the light.

Added.

(7) In Figure 4A, adult fish are schematized being placed into a 96-well plate. Suggest using the larval diagram as in Figure 6 for accuracy.

Done.

(8) In Figure 4, plot titles could be made more accessible, especially in 4 F. Suggest removing extraneous information / italicizing gene names, etc. In G, suggest writing out Baseline, Dark, and Light to make it more accessible. Same in 4B.

We have implemented some of the suggestions. In particular, italics were not used, since we are referring to the founder line, not the gene.

(9) Figure 6 legend B - after (barplots), suggest inserting the word "and", to make clear that barplots indicate host gene *and* closely related paralogs are indicated by dots.

Done.

(10) In methods: "To better capture all microexons..." This sentence is difficult to understand. Suggested edit: "we excluded *from our calculation?* tissues with known or expected partial overlap... from comparison (for example, ...).

Done.

(11) In the methods, "which were defined with similar parameters but -min_rep 2." Suggest spelling this out, e.g. "with similar parameters, but requiring sufficient read coverage in at least n=2 samples per valid tissue group, whereas we only required one.".

Done.

(12) RNA was extracted for event and knockout validations. What does event mean here?

Event refers to the validation of the exon regulatory pattern in WT tissues. We added this information.

Provide definitions for abbreviations:(1) (Figure 6) Delta corrected VST Expression.

Done.

(2) "Mic-hosting genes" paralogs.

Done.

(3) In Figure 1F, "emic" is not defined.

Done.

Misspellings:

All corrected.

(1) Figure 6B (percentile is spelled percentil).

(2) Figure 6B legend (bottom or top decile*).

(3) Figure 6D - Schizophrenia* genes.

(4) In Zebrafish husbandry and genotyping: suggest "srrm3 mutants grew more slowly.".

(5) In results, "reduced body size at 90pdf" > 90dpf.

**Reviewer #2 (Recommendations for the authors):**
(1) Characterization of microexon mutants (Figure 2): The semi-quantitative PCR with flanking primers (Figure 2, supplement1) is well-suited to assess successful deletion of the exon and enables detection of potential mis-splicing around the alternative segment. However, it does not quantify the impact on total transcript levels. The authors should complement those experiments with qPCR measures of the transcript levels - otherwise, it is difficult to link mutant phenotypes to isoforms (as opposed to alterations in the level of gene expression). This point is somewhat addressed in Figure 6 by the RNA Seq analysis but it might help to add data specifically in Figure 2.

As the Reviewer says, this point is explicitely addressed in Figure 6, where were show the change in the host gene's expression that follows the the removal of some microexons. We prefer to keep this in Figure 6, for consistency, as we believe this is not a direct (regulatory) consequence of the removal, but more likely a compensation effect.

(2) Social behavior alterations in juvenile fish: The authors report "increased leadership" in Srrm3 mutant fish. However, these fish have impaired vision. Thus, "increased leadership" may simply reflect the fact that they do not perceive their conspecifics and, thus, do not follow them. The heterozygous conspecific will then mostly follow the Srrm3 mutant which appears as the mutant exhibiting an increase in leadership. Figure 5D suggests that Srrm3 del and het fish have the same ratio of "neighbor in front" which would be consistent with the hypothesis that the change in this metric is a consequence of a loss of following behavior due to a loss of vision. The authors should either adjust the discussion of this point or assess with additional experiments whether this is indeed a "social phenotype" or rather a secondary consequence of a loss of vision.

The Reviewer is absolutely correct, and we have thus modified the short discussion directly related to these patterns.

(3) The discussion centers on potential reasons why only mild phenotypes are observed in the single microexon mutants. One caveat of the phenotypic analysis provided in the manuscript is that it does not very deeply explore the phenotypic space of neuronal morphologies or circuit function. The behavioral and anatomical read-outs are rather coarse. There are no experiments exploring fine-structure of neuronal projections in vivo or synapse number, morphology, or function. Moreover, no attempts are made to explore which cell types normally express the microexons to potentially focus the loss-of-function analysis to these specific cell types. Of course, such analysis would substantially expand the scope of a study that already covers a large number of mutant alleles. However, the authors may want to add a discussion of these limitations in the manuscript.

The Reviewer is correct. We aimed at covering this when referring to "(i) we may not be assessing the traits that these microexons are impacting, (ii) we may not have the sensitivity to robustly measure the magnitude of the changes caused by microexon removal". We have now added some of the specific points raised by the Reviewer as examples.

(4) Note typos in Figure 6D: "schizoFrenia", "WNT signIalling"

Done.

**Reviewer #3 (Recommendations for the authors):**
I only have a few minor suggestions for the authors.(1) It is interesting that a not insignificant number of microexon deletions (3/21) result in cryptic inclusions of intron fragments, and perhaps alludes to an as yet unreported molecular function of microexons in the regulation of host gene expression. Is it possible that microexon inclusion in these 3 genes could be important for expression? I think this requires some further discussion, as (if I'm not mistaken) microexons have thus far only been hypothesised to act as modulators of protein function, not as gene regulatory units.

While we see that microexon removal can impact expression of the host gene (Figure 6), this is likely a compensatory mechanism (or so we suggest). We do not think these three cases are related to a putative physiological regulation, since the cryptic exons appear only in the deletion line. On the contrary, we think these are "regulatory artifacts" that originate in the nonWT mutated context. I.e. we removed the exon but some splicing signals remained in the intron, which are then recoginized by the spliceosome that incorrectly includes a different piece of the intron.

(2) The flow of the text accompanying the molecular investigation of microexon function for evi5b and vav in Figure 3 could be improved. The text currently fades out with a speculative explanation for the lack of evi5b interaction phenotype. This final sentence could be moved to the discussion and replaced with a more general summary of the data.

We have now swapped the order in which these results are described and leave out the discussion about evi5b's microexon function.

(3) Is this a co-submission with Calhoun et al? If so, both papers should reference each other in the discussion and discuss the relative contributions of each.

Done

(4) "1 × 104 cells" in methods Nanobret paragraph should be superscript.

Done